# On the Convergence of Single-Call Stochastic Extra-Gradient Methods

**Yu-Guan Hsieh**
Univ. Grenoble Alpes, LJK and ENS Paris
38000 Grenoble, France.
yu-guan.hsieh@ens.fr

**Franck Iutzeler**
Univ. Grenoble Alpes, LJK
38000 Grenoble, France.
franck.iutzeler@univ-grenoble-alpes.fr

**Jérôme Malick**
CNRS, LJK
38000 Grenoble, France.
jerome.malick@univ-grenoble-alpes.fr

**Panayotis Mertikopoulos**
Univ. Grenoble Alpes, CNRS, Inria, Grenoble INP, LIG
38000 Grenoble, France.
panayotis.mertikopoulos@imag.fr

## Abstract

Variational inequalities have recently attracted considerable interest in machine learning as a flexible paradigm for models that go beyond ordinary loss function minimization (such as generative adversarial networks and related deep learning systems). In this setting, the optimal $\mathcal{O}(1/t)$ convergence rate for solving smooth monotone variational inequalities is achieved by the Extra-Gradient (EG) algorithm and its variants. Aiming to alleviate the cost of an extra gradient step per iteration (which can become quite substantial in deep learning applications), several algorithms have been proposed as surrogates to Extra-Gradient with a *single* oracle call per iteration. In this paper, we develop a synthetic view of such algorithms, and we complement the existing literature by showing that they retain a $\mathcal{O}(1/t)$ ergodic convergence rate in smooth, deterministic problems. Subsequently, beyond the monotone deterministic case, we also show that the last iterate of single-call, *stochastic* extra-gradient methods still enjoys a $\mathcal{O}(1/t)$ local convergence rate to solutions of *non-monotone* variational inequalities that satisfy a second-order sufficient condition.

## 1 Introduction

Deep learning is arguably the fastest-growing field in artificial intelligence: its applications range from image recognition and natural language processing to medical anomaly detection, drug discovery, and most fields where computers are required to make sense of massive amounts of data. In turn, this has spearheaded a prolific research thrust in optimization theory with the twofold aim of demystifying the successes of deep learning models and of providing novel methods to overcome their failures.

Introduced by Goodfellow et al. [20], generative adversarial networks (GANs) have become the youngest torchbearers of the deep learning revolution and have occupied the forefront of this drive in more ways than one. First, the adversarial training of deep neural nets has given rise to new challenges regarding the efficient allocation of parallelizable resources, the compatibility of the

|              | Lipschitz | | Lipschitz + Strong | |
|--------------|-----------|------------|-----------|-------------|
|              | Ergodic   | Last Iterate | Ergodic | Last Iterate |
| Deterministic | $\boxed{1/t}$ | Unknown | $1/t$ | $e^{-\rho t}$ [18, 25, 31] |
| Stochastic    | $1/\sqrt{t}$ [13, 18] | Unknown | $\boxed{1/t}$ | $\boxed{1/t}$ |

**Table 1:** The best known global convergence rates for single-call extra-gradient methods in monotone VI problems; logarithmic factors ignored throughout. A box indicates a contribution from this paper.

chosen architectures, etc. Second, the loss landscape in GANs is no longer that of a minimization problem but that of a zero-sum, min-max game – or, more generally, a *variational inequality* (VI).

Variational inequalities are a flexible and widely studied framework in optimization which, among others, incorporates minimization, saddle-point, Nash equilibrium, and fixed point problems. As such, there is an extensive literature devoted to solving variational inequalities in different contexts; for an introduction, see [4, 17] and references therein. In particular, in the setting of monotone variational inequalities with Lipschitz continuous operators, it is well known that the optimal rate of convergence is $\mathcal{O}(1/t)$, and that this rate is achieved by the Extra-Gradient (EG) algorithm of Korpelevich [23] and its Bregman variant, the Mirror-Prox (MP) algorithm of Nemirovski [32].[1]

These algorithms require two projections and two oracle calls per iteration, so they are more costly than standard Forward-Backward / descent methods. As a result, there are two complementary strands of literature aiming to reduce one (or both) of these cost multipliers – that is, the number of projections and/or the number of oracle calls per iteration. The first class contains algorithms like the Forward-Backward-Forward (FBF) method of Tseng [43], while the second focuses on gradient extrapolation mechanisms like Popov's modified Arrow–Hurwicz algorithm [37].

In deep learning, the latter direction has attracted considerably more interest than the former. The main reason for this is that neural net training often does not involve constraints (and, when it does, they are relatively cheap to handle). On the other hand, gradient calculations can become very costly, so a decrease in the number of oracle calls could offer significant practical benefits. In view of this, our aim in this paper is (*i*) to develop a synthetic approach to methods that retain the anticipatory properties of the Extra-Gradient algorithm while making a single oracle call per iteration; and (*ii*) to derive quantitative convergence results for such *single-call extra-gradient* (1-EG) algorithms.

**Our contributions.** Our first contribution complements the existing literature (reviewed below and in Section 3) by showing that the class of 1-EG algorithms under study attains the optimal $\mathcal{O}(1/t)$ convergence rate of the two-call method in deterministic variational inequalities with a monotone, Lipschitz continuous operator. Subsequently, we show that this rate is also achieved in *stochastic* variational inequalities with strongly monotone operators provided that the optimizer has access to an oracle with bounded variance (but not necessarily bounded second moments).

Importantly, this stochastic result concerns both the method's "ergodic average" (a weighted average of the sequence of points generated by the algorithm) as well as its "last iterate" (the last generated point). The reason for this dual focus is that averaging can be very useful in convex/monotone landscapes, but it is not as beneficial in non-monotone problems (where Jensen's inequality does not apply). On that account, last-iterate convergence results comprise an essential stepping stone for venturing beyond monotone problems.

Armed with these encouraging results, we then focus on *non-monotone* problems and show that, with high probability, the method's last iterate exhibits a $\mathcal{O}(1/t)$ local convergence rate to solutions of non-monotone variational inequalities that satisfy a second-order sufficient condition. To the best of our knowledge, this is the first convergence rate guarantee of this type for stochastic, non-monotone variational inequalities.

**Related work.** The prominence of Extra-Gradient/Mirror-Prox methods in solving variational inequalities and saddle-point problems has given rise to a vast corpus of literature which we cannot hope to do justice here. Especially in the context of adversarial networks, there has been a flurry

of recent activity relating variants of the Extra-Gradient algorithm to GAN training, see e.g., [9, 14, 18, 19, 24, 28, 44] and references therein. For concreteness, we focus here on algorithms with a single-call structure and refer the reader to Sections 3–5 for additional details.

The first variant of Extra-Gradient with a single oracle call per iteration dates back to Popov [37]. This algorithm was subsequently studied by, among others, Chiang et al. [10], Rakhlin and Sridharan [38, 39] and Gidel et al. [18]; see also [13, 25] for a "reflected" variant, [14, 30, 31, 36] for an "optimistic" one, and Section 3 for a discussion of the differences between these variants. In the context of deterministic, strongly monotone variational inequalities with Lipschitz continuous operators, the last iterate of the method was shown to exhibit a geometric convergence rate [18, 25, 31, 42]; similar geometric convergence results also extend to bilinear saddle-point problems [18, 36, 42], even though the operator involved is not strongly monotone. In turn, this implies the convergence of the method's ergodic average, but at a $\mathcal{O}(1/t)$ rate (because of the hysteresis of the average). In view of this, the fact that 1-EG methods retain the optimal $\mathcal{O}(1/t)$ convergence rate in deterministic variational inequalities without strong monotonicity assumptions closes an important gap in the literature.[2]

At the local level, the geometric convergence results discussed above echo a surge of interest in local convergence guarantees of optimization algorithms applied to games and saddle-point problems, see e.g., [1, 3, 15, 24] and references therein. In more detail, Liang and Stokes [24] proved local geometric convergence for several algorithms in possibly non-monotone saddle-point problems under a local smoothness condition. In a similar vein, Daskalakis and Panageas [15] analyzed the limit points of (optimistic) gradient descent, and showed that local saddle points are stable stationary points; subsequently, Adolphs et al. [1] and Mazumdar et al. [27] proposed a class of algorithms that eliminate stationary points which are not local Nash equilibria.

Geometric convergence results of this type are inherently deterministic because they rely on an associated resolvent operator being firmly nonexpansive – or, equivalently, rely on the use of the center manifold theorem. In a stochastic setting, these techniques are no longer applicable because the contraction property cannot be maintained in the presence of noise; in fact, unless the problem at hand is amenable to variance reduction – e.g., as in [6, 9, 21] – geometric convergence is not possible if the noise process is even weakly isotropic. Instead, for monotone problems, Cui and Shanbhag [13] and Gidel et al. [18] showed that the ergodic average of the method attains a $\mathcal{O}(1/\sqrt{t})$ convergence rate. Our global convergence results for stochastic variational inequalities improve this rate to $\mathcal{O}(1/t)$ in strongly monotone variational inequalities for both the method's ergodic average and its last iterate. In the same light, our local $\mathcal{O}(1/t)$ convergence results for *non-monotone* variational inequalities provide a key extension of local, deterministic convergence results to a fully stochastic setting, all the while retaining the fastest convergence rate for monotone variational inequalities.

For convenience, our contributions relative to the state of the art are summarized in Table 1.

## 2 Problem setup and blanket assumptions

**Variational inequalities.** We begin by presenting the basic variational inequality framework that we will consider throughout the sequel. To that end, let $\mathcal{X}$ be a nonempty closed convex subset of $\mathbb{R}^d$, and let $V \colon \mathbb{R}^d \to \mathbb{R}^d$ be a single-valued operator on $\mathbb{R}^d$. In its most general form, the *variational inequality* (VI) problem associated to $V$ and $\mathcal{X}$ can be stated as:

$$\text{Find } x^\star \in \mathcal{X} \text{ such that } \langle V(x^\star), x - x^\star \rangle \geq 0 \text{ for all } x \in \mathcal{X}. \tag{VI}$$

To provide some intuition about (VI), we discuss two important examples below:

**Example 1** (Loss minimization). Suppose that $V = \nabla f$ for some smooth loss function $f$ on $\mathcal{X} = \mathbb{R}^d$. Then, $x^\star \in \mathcal{X}$ is a solution to (VI) if and only if $\nabla f(x^\star) = 0$, i.e., if and only if $x^\star$ is a critical point of $f$. Of course, if $f$ is convex, any such solution is a global minimizer. □

**Example 2** (Min-max optimization). Suppose that $\mathcal{X}$ decomposes as $\mathcal{X} = \Theta \times \Phi$ with $\Theta = \mathbb{R}^{d_1}$, $\Phi = \mathbb{R}^{d_2}$, and assume $V = (\nabla_\theta \mathcal{L}, -\nabla_\phi \mathcal{L})$ for some smooth function $\mathcal{L}(\theta, \phi)$, $\theta \in \Theta, \phi \in \Phi$. As in

Example 1 above, the solutions to (VI) correspond to the critical points of $\mathcal{L}$; if, in addition, $\mathcal{L}$ is convex-concave, any solution $x^\star = (\theta^\star, \phi^\star)$ of (VI) is a global *saddle-point*, i.e.,

$$\mathcal{L}(\theta^\star, \phi) \leq \mathcal{L}(\theta^\star, \phi^\star) \leq \mathcal{L}(\theta, \phi^\star) \quad \text{for all } \theta \in \Theta \text{ and all } \phi \in \Phi.$$

Given the original formulation of GANs as (stochastic) saddle-point problems [20], this observation has been at the core of a vigorous literature at the interface between optimization, game theory, and deep learning, see e.g., [9, 14, 18, 24, 28, 36, 44] and references therein. □

The operator analogue of convexity for a function is *monotonicity*, i.e.,

$$\langle V(x') - V(x), x' - x \rangle \geq 0 \quad \text{for all } x, x' \in \mathbb{R}^d.$$

Specifically, when $V = \nabla f$ for some sufficiently smooth function $f$, this condition is equivalent to $f$ being convex [4]. In this case, following Nesterov [34, 35] and Juditsky et al. [22], the quality of a candidate solution $\hat{x} \in \mathcal{X}$ can be assessed via the so-called *error* (or *merit*) *function*

$$\mathrm{Err}(\hat{x}) = \sup_{x \in \mathcal{X}} \langle V(x), \hat{x} - x \rangle$$

and/or its restricted variant

$$\mathrm{Err}_R(\hat{x}) = \max_{x \in \mathcal{X}_R} \langle V(x), \hat{x} - x \rangle,$$

where $\mathcal{X}_R \equiv \mathcal{X} \cap \mathbb{B}_R(0) = \{x \in \mathcal{X} : \|x\| \leq R\}$ denotes the "restricted domain" of the problem. More precisely, we have the following basic result.

**Lemma 1** (Nesterov, 2007). *Assume $V$ is monotone. If $x^\star$ is a solution of* (VI)*, we have* $\mathrm{Err}(x^\star) = 0$ *and* $\mathrm{Err}_R(x^\star) = 0$ *for all sufficiently large $R$. Conversely, if* $\mathrm{Err}_R(\hat{x}) = 0$ *for large enough $R > 0$ and some $\hat{x} \in \mathcal{X}_R$, then $\hat{x}$ is a solution of* (VI)*.*

In light of this result, $\mathrm{Err}$ and $\mathrm{Err}_R$ will be among our principal measures of convergence in the sequel.

**Blanket assumptions.** With all this in hand, we present below the main assumptions that will underlie the bulk of the analysis to follow.

**Assumption 1.** The solution set $\mathcal{X}^\star$ of (VI) is nonempty.

**Assumption 2.** The operator $V$ is $\beta$-Lipschitz continuous, i.e.,

$$\|V(x') - V(x)\| \leq \beta \|x' - x\| \quad \text{for all } x, x' \in \mathbb{R}^d.$$

**Assumption 3.** The operator $V$ is monotone.

In some cases, we will also strengthen Assumption 3 to:

**Assumption 3(s).** The operator $V$ is $\alpha$-strongly monotone, i.e.,

$$\langle V(x') - V(x), x' - x \rangle \geq \alpha \|x' - x\|^2 \quad \text{for some } \alpha > 0 \text{ and all } x, x' \in \mathbb{R}^d.$$

Throughout our paper, we will be interested in sequences of points $X_t \in \mathcal{X}$ generated by algorithms that can access the operator $V$ via a *stochastic oracle* [33].[3] Formally, this is a black-box mechanism which, when called at $X_t \in \mathcal{X}$, returns the estimate

$$V_t = V(X_t) + Z_t, \tag{1}$$

where $Z_t \in \mathbb{R}^d$ is an additive noise variable satisfying the following hypotheses:

    *a) Zero-mean:*      $\mathbb{E}[Z_t \mid \mathcal{F}_t] = 0.$

    *b) Finite variance:*    $\mathbb{E}[\|Z_t\|^2 \mid \mathcal{F}_t] \leq \sigma^2.$

In the above, $\mathcal{F}_t$ denotes the history (natural filtration) of $X_t$, so $X_t$ is adapted to $\mathcal{F}_t$ by definition; on the other hand, since the $t$-th instance of $Z_t$ is generated randomly from $X_t$, $Z_t$ is *not* adapted to $\mathcal{F}_t$. Obviously, if $\sigma^2 = 0$, we have the deterministic, *perfect feedback* case $V_t = V(X_t)$.

## 3 Algorithms

**The Extra-Gradient algorithm.**    In the general framework outlined in the previous section, the Extra-Gradient (EG) algorithm of Korpelevich [23] can be stated in recursive form as

$$
\begin{aligned}
X_{t+1/2} &= \Pi_{\mathcal{X}}(X_t - \gamma_t V_t) \\
X_{t+1} &= \Pi_{\mathcal{X}}(X_t - \gamma_t V_{t+1/2})
\end{aligned}
\tag{EG}
$$

where $\Pi_{\mathcal{X}}(y) := \arg\min_{x \in \mathcal{X}} \|y - x\|$ denotes the Euclidean projection of $y \in \mathbb{R}^d$ onto the closed convex set $\mathcal{X}$ and $\gamma_t > 0$ is a variable step-size sequence. Using this formulation as a starting point, the main idea behind the method can be described as follows: at each $t = 1, 2, \dots$, the oracle is called at the algorithm's current – or *base* – state $X_t$ to generate an intermediate – or *leading* – state $X_{t+1/2}$; subsequently, the base state $X_t$ is updated to $X_{t+1}$ using gradient information from the leading state $X_{t+1/2}$, and the process repeats. Heuristically, the extra oracle call allows the algorithm to "anticipate" the landscape of $V$ and, in so doing, to achieve improved convergence results relative to standard projected gradient / forward-backward methods; for a detailed discussion, we refer the reader to [7, 17] and references therein.

**Single-call variants of the Extra-Gradient algorithm.**    Given the significant computational overhead of gradient calculations, a key desideratum is to drop the second oracle call in (EG) while retaining the algorithm's "anticipatory" properties. In light of this, we will focus on methods that perform a *single* oracle call at the leading state $X_{t+1/2}$, but replace the update rule for $X_{t+1/2}$ (and, possibly, $X_t$ as well) with a proxy that compensates for the missing gradient. Concretely, we will examine the following family of *single-call extra-gradient* (1-EG) algorithms:

1. *Past Extra-Gradient* (PEG) [10, 18, 37]:

$$
\begin{aligned}
X_{t+1/2} &= \Pi_{\mathcal{X}}(X_t - \gamma_t V_{t-1/2}) \\
X_{t+1} &= \Pi_{\mathcal{X}}(X_t - \gamma_t V_{t+1/2})
\end{aligned}
\tag{PEG}
$$

   [Proxy: use $V_{t-1/2}$ instead of $V_t$ in the calculation of $X_{t+1/2}$]

2. *Reflected Gradient* (RG) [8, 13, 25]:

$$
\begin{aligned}
X_{t+1/2} &= X_t - (X_{t-1} - X_t) \\
X_{t+1} &= \Pi_{\mathcal{X}}(X_t - \gamma_t V_{t+1/2})
\end{aligned}
\tag{RG}
$$

   [Proxy: use $(X_{t-1} - X_t)/\gamma_t$ instead of $V_t$ in the calculation of $X_{t+1/2}$; no projection]

3. *Optimistic Gradient* (OG) [14, 30, 31, 36]:

$$
\begin{aligned}
X_{t+1/2} &= \Pi_{\mathcal{X}}(X_t - \gamma_t V_{t-1/2}) \\
X_{t+1} &= X_{t+1/2} + \gamma_t V_{t-1/2} - \gamma_t V_{t+1/2}
\end{aligned}
\tag{OG}
$$

   [Proxy: use $V_{t-1/2}$ instead of $V_t$ in the calculation of $X_{t+1/2}$; use $X_{t+1/2} + \gamma_t V_{t-1/2}$ instead of $X_t$ in the calculation of $X_{t+1}$; no projection]

These are the main algorithmic schemes that we will consider, so a few remarks are in order. First, given the extensive literature on the subject, this list is not exhaustive; see e.g., [30, 31, 36] for a generalization of (OG), [26] for a variant that employs averaging to update the algorithm's base state $X_t$, and [19] for a proxy defined via "negative momentum". Nevertheless, the algorithms presented above appear to be the most widely used single-call variants of (EG), and they illustrate very clearly the two principal mechanisms for approximating missing gradients: (*i*) using past gradients (as in the PEG and OG variants); and/or (*ii*) using a difference of successive states (as in the RG variant).

We also take this opportunity to provide some background and clear up some issues on terminology regarding the methods presented above. First, the idea of using past gradients dates back at least to Popov [37], who introduced (PEG) as a "modified Arrow–Hurwicz" method a few years after the original paper of Korpelevich [23]; the same algorithm is called "meta" in [10] and "extrapolation from the past" in [18] (but see also the note regarding optimism below). The terminology "Reflected

Gradient" and the precise formulation that we use here for (RG) is due to Malitsky [25]. The well-known primal-dual algorithm of Chambolle and Pock [8] can be seen as a one-sided, alternating variant of the method for saddle-point problems; see also [44] for a more recent take.

Finally, the terminology "optimistic" is due to Rakhlin and Sridharan [38, 39], who provided a unified view of (PEG) and (EG) based on the sequence of oracle vectors used to update the algorithm's leading state $X_{t+1/2}$.[4] Because the framework of [38, 39] encompasses two different algorithms, there is some danger of confusion regarding the use of the term "optimism"; in particular, both (EG) and (PEG) can be seen as instances of optimism. The specific formulation of (OG) that we present here is the projected version of the algorithm considered by Daskalakis et al. [14];[5] by contrast, the "optimistic" method of Mertikopoulos et al. [28] is equivalent to (EG) – not (PEG) or (OG).

The above shows that there can be a broad array of single-call extra-gradients methods depending on the specific proxy used to estimate the missing gradient, whether it is applied to the algorithm's base or leading state, when (or where) a projection operator is applied, etc. The contact point of all these algorithms is the unconstrained setting ($\mathcal{X} = \mathbb{R}^d$) where they are exactly equivalent:

**Proposition 1.** *Suppose that the* 1*-EG methods presented above share the same initialization,* $X_0 = X_1 \in \mathcal{X}$, $V_{1/2} = 0$, *and are run with the same, constant step-size* $\gamma_t \equiv \gamma$ *for all* $t \geq 1$. *If* $\mathcal{X} = \mathbb{R}^d$, *the generated iterates* $X_t$ *coincide for all* $t \geq 1$.

The proof of this proposition follows by a simple rearrangement of the update rules for (PEG), (RG) and (OG), so we omit it. In the projected case, the 1-EG updates presented above are no longer equivalent – though, of course, they remain closely related.

## 4  Deterministic analysis

We begin with the deterministic analysis, i.e., when the optimizer receives oracle feedback of the form (1) with $\sigma = 0$. In terms of presentation, we keep the global and local cases separated and we interleave our results for the generated sequence $X_t$ and its *ergodic average*. To streamline our presentation, we defer the details of the proofs to the paper's supplement and only discuss here the main ideas.

### 4.1  Global convergence

Our first result below shows that the algorithms under study achieve the optimal $\mathcal{O}(1/t)$ ergodic convergence rate in monotone problems with Lipschitz continuous operators.

**Theorem 1.** *Suppose that* $V$ *satisfies Assumptions 1–3. Assume further that a* 1*-EG algorithm is run with perfect oracle feedback and a constant step-size* $\gamma < 1/(c\beta)$, *where* $c = 1 + \sqrt{2}$ *for the RG variant and* $c = 2$ *for the PEG and OG variants. Then, for all* $R > 0$, *we have*

$$\operatorname{Err}_R\left(\bar{X}_t\right) \leq \frac{R^2 + \|X_1 - X_{1/2}\|^2}{2\gamma t}$$

*where* $\bar{X}_t = t^{-1}\sum_{s=1}^{t} X_{s+1/2}$ *is the ergodic average of the algorithm's sequence of leading states.*

This result shows that the EG and 1-EG algorithms share the same convergence rate guarantees, so we can safely drop one gradient calculation per iteration in the monotone case. The proof of the theorem is based on the following technical lemma which enables us to treat the different variants of the 1-EG method in a unified way.

**Lemma 2.** *Assume that* $V$ *satisfies Assumption 3 (monotonicity). Suppose further that the sequence* $(X_t)_{t\in\mathbb{N}/2}$ *of points in* $\mathbb{R}^d$ *satisfies the following "quasi-descent" inequality with* $\mu_s, \lambda_s \geq 0$:

$$\|X_{s+1} - p\|^2 \leq \|X_s - p\|^2 - 2\lambda_s\langle V(X_{s+1/2}), X_{s+1/2} - p\rangle + \mu_s - \mu_{s+1} \tag{3}$$

*for all $p \in \mathcal{X}_R$ and all $s \in \{1, \ldots, t\}$. Then,*

$$\mathrm{Err}_R \left( \frac{\sum_{s=1}^{t} \lambda_s X_{s+1/2}}{\sum_{s=1}^{t} \lambda_s} \right) \leq \frac{R^2 + \mu_1}{2 \sum_{s=1}^{t} \lambda_s}.$$

*Remark* 1. For Examples 1 and 2 it is possible to state both Theorem 1 and Lemma 2 with more adapted measures. We refer the readers to the supplement for more details.

The use of Lemma 2 is tailored to time-averaged sequences like $\bar{X}_t$, and relies on establishing a suitable "quasi-descent inequality" of the form (3) for the iterates of 1-EG. Doing this requires in turn a careful comparison of successive iterates of the algorithm via the Lipschitz continuity assumption for $V$; we defer the precise treatment of this argument to the paper's supplement.

On the other hand, because the role of averaging is essential in this argument, the convergence of the algorithm's last iterate requires significantly different techniques. To the best of our knowledge, there are no comparable convergence rate guarantees for $X_t$ under Assumptions 1–3; however, if Assumption 3 is strengthened to Assumption 3(s), the convergence of $X_t$ to the (necessarily unique) solution of (VI) occurs at a geometric rate. For completeness, we state here a consolidated version of the geometric convergence results of Malitsky [25], Gidel et al. [18], and Mokhtari et al. [31].

**Theorem 2.** *Assume that $V$ satisfies Assumptions 1, 2 and 3(s), and let $x^\star$ denote the (necessarily unique) solution of (VI). If a 1-EG algorithm is run with a sufficiently small step-size $\gamma$, the generated sequence $X_t$ converges to $x^\star$ at a rate of $\|X_t - x^\star\| = \mathcal{O}(\exp(-\rho t))$ for some $\rho > 0$.*

## 4.2 Local convergence

We continue by presenting a local convergence result for deterministic, *non-monotone* problems. To state it, we will employ the following notion of regularity in lieu of Assumptions 1–3 and 3(s).

**Definition 3.** We say that $x^\star$ is a *regular solution* of (VI) if $V$ is $C^1$-smooth in a neighborhood of $x^\star$ and the Jacobian $\mathrm{Jac}_V(x^\star)$ is positive-definite along rays emanating from $x^\star$, i.e.,

$$z^\top \mathrm{Jac}_V(x^\star)z \equiv \sum_{i,j=1}^{d} z_i \frac{\partial V_i}{\partial x_j}(x^\star)z_j > 0$$

for all $z \in \mathbb{R}^d \setminus \{0\}$ that are tangent to $\mathcal{X}$ at $x^\star$.

This notion of regularity is an extension of similar conditions that have been employed in the local analysis of loss minimization and saddle-point problems. More precisely, if $V = \nabla f$ for some loss function $f$, this definition is equivalent to positive-definiteness of the Hessian along qualified constraints [5, Chap. 3.2]. As for saddle-point problems and smooth games, variants of this condition can be found in several different sources, see e.g., [16, 24, 29, 40, 41] and references therein.

Under this condition, we obtain the following local geometric convergence result for 1-EG methods.

**Theorem 4.** *Let $x^\star$ be a regular solution of (VI). If a 1-EG method is run with perfect oracle feedback and is initialized sufficiently close to $x^\star$ with a sufficiently small constant step-size, we have $\|X_t - x^\star\| = \mathcal{O}(\exp(-\rho t))$ for some $\rho > 0$.*

The proof of this theorem relies on showing that (*i*) $V$ essentially behaves like a smooth, strongly monotone operator close to $x^\star$; and (*ii*) if the method is initialized in a small enough neighborhood of $x^\star$, it will remain in said neighborhood for all $t$. As a result, Theorem 4 essentially follows by "localizing" Theorem 2 to this neighborhood.

As a preamble to our stochastic analysis in the next section, we should state here that, albeit straightforward, the proof strategy outlined above breaks down if we have access to $V$ only via a *stochastic* oracle. In this case, a single "bad" realization of the feedback noise $Z_t$ could drive the process away from the attraction region of any local solution of (VI). For this reason, the stochastic analysis requires significantly different tools and techniques and is considerably more intricate.

# 5 Stochastic analysis

We now present our analysis for stochastic variational inequalities with oracle feedback of the form (1). For concreteness, given that the PEG variant of the 1-EG method employs the most straightforward

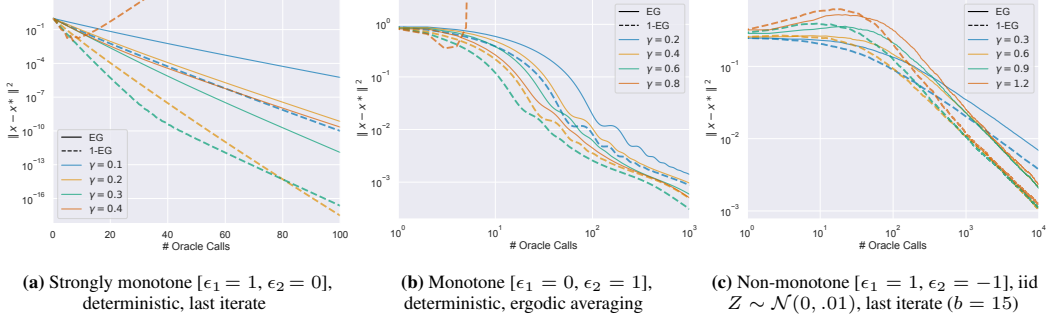

**(a)** Strongly monotone $[\epsilon_1 = 1, \epsilon_2 = 0]$, deterministic, last iterate

**(b)** Monotone $[\epsilon_1 = 0, \epsilon_2 = 1]$, deterministic, ergodic averaging

**(c)** Non-monotone $[\epsilon_1 = 1, \epsilon_2 = -1]$, iid $Z \sim \mathcal{N}(0, .01)$, last iterate $(b = 15)$

**Figure 1:** Illustration of the performance of EG and 1-EG in the (a priori non-monotone) saddle-point problem

$$\mathcal{L}(\theta, \phi) = 2\epsilon_1 \theta^\top A_1 \theta + \epsilon_2 \left(\theta^\top A_2 \theta\right)^2 - 2\epsilon_1 \phi^\top B_1 \phi - \epsilon_2 \left(\phi^\top B_2 \phi\right)^2 + 4\theta^\top C\phi$$

on the full unconstrained space $\mathcal{X} = \mathbb{R}^d = \mathbb{R}^{d_1 \times d_2}$ with $d_1 = d_2 = 1000$ and $A_1, B_1, A_2, B_2 \succ 0$. We choose three situations representative of the settings considered in the paper: (*a*) linear convergence of the last iterate of the deterministic methods in strongly monotone problems; (*b*) the $\mathcal{O}(1/t)$ convergence of the ergodic average in monotone, deterministic problems; and (*c*) the $\mathcal{O}(1/t)$ local convergence rate of the method's last iterate in stochastic, *non-monotone* problems. For (*a*) and (*b*), the origin is the unique solution of (VI), and for (*c*) it is a regular solution thereof. We observe that 1-EG consistently outperforms EG in terms of oracle calls for a fixed step-size, and the observed rates are consistent with the rates reported in Table 1.

proxy mechanism, we will focus on this variant throughout; for the other variants, the proofs and corresponding explicit expressions follow from the same rationale (as in the case of Theorem 1).

## 5.1 Global convergence

As we mentioned in the introduction, under Assumptions 1–3, Cui and Shanbhag [13] and Gidel et al. [18] showed that 1-EG methods attain a $\mathcal{O}(1/\sqrt{t})$ ergodic convergence rate. By strengthening Assumption 3 to Assumption 3(s), we show that this result can be augmented in two synergistic ways: under Assumptions 1, 2 and 3(s), both the last iterate and the ergodic average of 1-EG achieve a $\mathcal{O}(1/t)$ convergence rate.

**Theorem 5.** *Suppose that $V$ satisfies Assumptions 1, 2 and 3(s), and assume that (PEG) is run with stochastic oracle feedback of the form* (1) *and a step-size of the form $\gamma_t = \gamma/(t + b)$ for some $\gamma > 1/\alpha$ and $b \geq 4\beta\gamma$. Then, the generated sequence of the algorithm's base states satisfies*

$$\mathbb{E}[\|X_t - x^\star\|^2] \leq \frac{6\gamma^2\sigma^2}{\alpha\gamma - 1}\frac{1}{t} + o\left(\frac{1}{t}\right),$$

*while its ergodic average $\bar{X}_t = t^{-1}\sum_{s=1}^t X_s$ enjoys the bound*

$$\mathbb{E}[\|\bar{X}_t - x^\star\|^2] \leq \frac{6\gamma^2\sigma^2}{\alpha\gamma - 1}\frac{\log t}{t} + o\left(\frac{\log t}{t}\right).$$

Regarding our proof strategy for the last iterate of the process, we can no longer rely either on a contraction argument or the averaging mechanism that yields the $\mathcal{O}(1/\sqrt{t})$ ergodic convergence rate. Instead, we show in the appendix that $X_t$ is (stochastically) quasi-Fejér in the sense of [11, 12]; then, leveraging the method's specific step-size, we employ successive numerical sequence estimates to control the summability error and obtain the $\mathcal{O}(1/t)$ rate.

## 5.2 Local convergence

We proceed to examine the convergence of the method in the stochastic, *non-monotone* case. Our main result in this regard is the following.

**Theorem 6.** *Let $x^\star$ be a regular solution of* (VI) *and fix a tolerance level $\delta > 0$. Suppose further that (PEG) is run with stochastic oracle feedback of the form* (1) *and a variable step-size of the form $\gamma_t = \gamma/(t + b)$ for some $\gamma > 1/\alpha$ and large enough $b$. Then:*

(a) *There are neighborhoods $U$ and $U_1$ of $x^\star$ in $\mathcal{X}$ such that, if $X_{1/2} \in U, X_1 \in U_1$, the event*

$$E_\infty = \{X_{t+1/2} \in U \text{ for all } t = 1, 2, \dots\}$$

*occurs with probability at least $1 - \delta$.*

(b) *Conditioning on the above, we have:*

$$\mathbb{E}[\|X_t - x^\star\|^2 \mid E_\infty] \leq \frac{4\gamma^2(M^2 + \sigma^2)}{(\alpha\gamma - 1)(1 - \delta)} \frac{1}{t} + o\left(\frac{1}{t}\right),$$

*where $M = \sup_{x \in U} \|V(x)\| < \infty$ and $\alpha = \inf_{x \in U} \langle V(x), x - x^\star \rangle / \|x - x^\star\|^2 > 0$.*

The finiteness of $M$ and the positivity of $\alpha$ are both consequences of the regularity of $x^\star$ and their values only depend on the size of the neighborhood $U$. Taking a larger $U$ would increase the algorithm's certified initialization basin but it would also negatively impact its convergence rate (since $M$ would increase while $\alpha$ would decrease). Likewise, the neighborhood $U_1$ only depends on the size of $U$ and, as we explain in the appendix, it suffices to take $U_1$ to be "one fourth" of $U$.

From the above, it becomes clear that the situation is significantly more involved than the corresponding deterministic analysis. This is also reflected in the proof of Theorem 6 which requires completely new techniques, well beyond the straightforward localization scheme underlying Theorem 4. More precisely, a key step in the proof (which we detail in the appendix) is to show that the iterates of the method remain close to $x^\star$ for all $t$ with arbitrarily high probability. In turn, this requires showing that the probability of getting a string of "bad" noise realizations of arbitrary length is controllably small. Even then however, the global analysis *still* cannot be localized because conditioning changes the probability law under which the oracle noise is unbiased. Accounting for this conditional bias requires a surprisingly delicate probabilistic argument which we also detail in the supplement.

## 6   Concluding remarks

Our aim in this paper was to provide a synthetic view of single-call surrogates to the Extra-Gradient algorithm, and to establish optimal convergence rates in a range of different settings – deterministic, stochastic, and/or non-monotone. Several interesting avenues open up as a result, from extending the theory to more general Bregman proximal settings, to developing an adaptive version as in the recent work [2] for two-call methods. We defer these research directions to future work.

## Acknowledgments

This work benefited from financial support by MIAI Grenoble Alpes (Multidisciplinary Institute in Artificial Intelligence). P. Mertikopoulos was partially supported by the French National Research Agency (ANR) grant ORACLESS (ANR–16–CE33–0004–01) and the EU COST Action CA16228 "European Network for Game Theory" (GAMENET).

## Footnotes

[1]Korpelevich [23] proved the method's asymptotic convergence for pseudomonotone variational inequalities. The $\mathcal{O}(1/t)$ convergence rate was later established by Nemirovski [32] with ergodic averaging.

[2]A few weeks after the submission of our paper, we were made aware of a very recent preprint by Mokhtari et al. [30] which also establishes a $\mathcal{O}(1/t)$ convergence rate for the algorithm's "optimistic" variant in saddle-point problems (in terms of the Nikaido–Isoda gap function). To the best of our knowledge, this is the closest result to our own in the literature.

[3]Depending on the algorithm, the sequence index $t$ may take positive integer or half-integer values (or both).

[4]More precisely, Rakhlin and Sridharan [38, 39] use the term Optimistic Mirror Descent (OMD) in reference to the Mirror-Prox method of Nemirovski [32], itself a variant of (EG) with projections defined by means of a Bregman function; for a related treatment, see Nesterov [34] and Juditsky et al. [22].

[5]To see this, note that the difference between two consecutive intermediate steps $X_{t-1/2}$ and $X_{t+1/2}$ can be written as $X_{t+1/2} = \Pi_{\mathcal{X}}(X_{t-1/2} - (\gamma_{t-1} + \gamma_t)V_{t-1/2} + \gamma_{t-1}V_{t-3/2})$. Writing (OG) in the form presented above shows that (OG) can also be viewed as a single-call variant of the FBF method of Tseng [43].

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
