[Supplementary Material]

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

[6]Please refer to the proof of Theorem 1 for the exact value of $\mu_t$.

[7]In particular this also holds for RG since then $X_{t+\frac{1}{2}} - X_t = X_t - X_{t-1} = \Pi_{\mathcal{X}}(X_{t-1} - \gamma V(X_{t-\frac{1}{2}})) - \Pi_{\mathcal{X}}(X_{t-1})$.

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

## A  Technical lemmas

**Lemma A.1.** *Let* $x, y \in \mathbb{R}^d$ *and* $\mathcal{C} \subseteq \mathbb{R}^d$ *be a closed convex set. We set* $x^+ := \Pi_{\mathcal{C}}(x - y)$. *For all* $p \in \mathcal{C}$, *we have*

$$\|x^+ - p\|^2 \leq \|x - p\|^2 - 2\langle y, x^+ - p\rangle - \|x^+ - x\|^2.$$

*Proof.* Since $p \in \mathcal{C}$, we have the following property $\langle x^+ - (x - y), x^+ - p\rangle \leq 0$, leading to

$$
\begin{aligned}
\|x^+ - p\|^2 &= \|x^+ - x + x - p\|^2 \\
&= \|x - p\|^2 + 2\langle x^+ - x, x - p\rangle + \|x^+ - x\|^2 \\
&= \|x - p\|^2 + 2\langle x^+ - x, x^+ - p\rangle - \|x^+ - x\|^2 \\
&\leq \|x - p\|^2 - 2\langle y, x^+ - p\rangle - \|x^+ - x\|^2. \qquad \square
\end{aligned}
$$

**Lemma A.2.** *Let* $x, y_1, y_2 \in \mathbb{R}^d$ *and* $\mathcal{C}_1, \mathcal{C}_2 \subseteq \mathbb{R}^d$ *be two closed convex sets. We set* $x_1^+ := \Pi_{\mathcal{C}_1}(x - y_1)$ *and* $x_2^+ := \Pi_{\mathcal{C}_2}(x - y_2)$.

(a) *If* $\mathcal{C}_2 = \mathbb{R}^d$, *for all* $p \in \mathbb{R}^d$, *it holds*

$$\|x_2^+ - p\|^2 = \|x - p\|^2 - 2\langle y_2, x_1^+ - p\rangle + \|x_2^+ - x_1^+\|^2 - \|x_1^+ - x\|^2.$$

(b) *If* $\mathcal{C}_2 \subseteq \mathcal{C}_1$, *for all* $p \in \mathcal{C}_2$, *it holds*

$$
\begin{aligned}
\|x_2^+ - p\|^2 &\leq \|x - p\|^2 - 2\langle y_2, x_1^+ - p\rangle + 2\langle y_2 - y_1, x_1^+ - x_2^+\rangle \\
&\quad - \|x_2^+ - x_1^+\|^2 - \|x_1^+ - x\|^2 \\
&\leq \|x - p\|^2 - 2\langle y_2, x_1^+ - p\rangle + \|y_2 - y_1\|^2 - \|x_1^+ - x\|^2. \tag{A.1}
\end{aligned}
$$

*Proof.* (a) We develop

$$
\begin{aligned}
\|x_2^+ - p\|^2 &= \|x_2^+ - x_1^+ + x_1^+ - x + x - p\|^2 \\
&= \|x_2^+ - x_1^+\|^2 + \|x_1^+ - x\|^2 + \|x - p\|^2 \\
&\quad + 2\langle x_2^+ - x_1^+, x_1^+ - p\rangle + 2\langle x^+ - x, x - p\rangle \\
&= \|x_2^+ - x_1^+\|^2 - \|x_1^+ - x\|^2 + \|x - p\|^2 \\
&\quad + 2\langle x_2^+ - x_1^+, x_1^+ - p\rangle + 2\langle x_1^+ - x, x_1^+ - p\rangle \\
&= \|x - p\|^2 - 2\langle y_2, x_1^+ - p\rangle + \|x_2^+ - x_1^+\|^2 - \|x_1^+ - x\|^2,
\end{aligned}
$$

where in the last line we use $x_2^+ - x = -y_2$ since $\mathcal{C}_2 = \mathbb{R}^d$.

(b) With $x_2^+ \in \mathcal{C}_2 \subseteq \mathcal{C}_1$, we can apply Lemma A.1 to $(x, y, x^+, p, \mathcal{C}) \leftarrow (x, y_2, x_2^+, p, \mathcal{C}_2)$ and $(x, y, x^+, p, \mathcal{C}) \leftarrow (x, y_1, x_1^+, x_2^+, \mathcal{C}_1)$, which yields

$$\|x_2^+ - p\|^2 \leq \|x - p\|^2 - 2\langle y_2, x_2^+ - p\rangle - \|x_2^+ - x\|^2, \tag{A.2}$$
$$\|x_1^+ - x_2^+\|^2 \leq \|x - x_2^+\|^2 - 2\langle y_1, x_1^+ - x_2^+\rangle - \|x_1^+ - x\|^2. \tag{A.3}$$

By summing (A.2) and (A.3), we readily get the first inequality of (A.1). We conclude with help of Young's inequality $2\langle y_2 - y_1, x_1^+ - x_2^+\rangle \leq \|y_2 - y_1\|^2 + \|x_1^+ - x_2^+\|^2$. $\qquad \square$

**Lemma A.3** (Chung [11, Lemma 1]). *Let* $(a_t)_{t\in\mathbb{N}}$ *be a sequence of real numbers and* $b, t_0 \in \mathbb{N}$ *such that for all* $t \geq t_0$,

$$a_{t+1} \leq \left(1 - \frac{q}{t + b}\right) a_t + \frac{q'}{(t + b)^2}, \tag{A.4}$$

*where* $q > 1$ *and* $q' > 0$. *Then,*

$$a_t \leq \frac{q'}{q - 1}\frac{1}{t} + o\left(\frac{1}{t}\right). \tag{A.5}$$

*Proof.* For the sake of completeness, we provide a basic proof for the above lemma (which is a direct corollary of Chung [11, Lemma 1]). Let $q > 1$ and $k \in \mathbb{N}$, we have

$$\frac{1}{k+1} - \left(1 - \frac{q}{k}\right)\frac{1}{k} = \frac{q}{k^2} - \left(\frac{1}{k} - \frac{1}{k+1}\right) = \frac{q-1}{k^2} + \frac{1}{k^2(k+1)}.$$

This shows that for any $q' > 0$

$$\frac{q'}{q-1}\left(\frac{1}{k+1} - \left(1 - \frac{q}{k}\right)\frac{1}{k}\right) = \frac{q'}{k^2} + \frac{q'}{k^2(k+1)(q-1)} \geq \frac{q'}{k^2}. \tag{A.6}$$

By substituting $k \leftarrow t + b$, (A.4) combined with (A.6) yields

$$a_{t+1} - \frac{q'}{q-1}\frac{1}{t+b+1} \leq \left(1 - \frac{q}{t+b}\right)\left(a_t - \frac{q'}{q-1}\frac{1}{t+b}\right). \tag{A.7}$$

Let us define $a_t' := a_t - q'/((q-1)(t+b))$. (A.7) becomes

$$a_{t+1}' \leq \left(1 - \frac{q}{t+b}\right)a_t'. \tag{A.8}$$

This inequality holds for all $t \geq t_0$. Then, either:
• $a_t'$ becomes non-positive for some $t > t_1 = \max(t_0, \lfloor q \rfloor - b)$, and (A.8) implies that this is also the case for all subsequent $t$, which leads to

$$a_t \leq \frac{q'}{q-1}\frac{1}{t+b}.$$

• or $a_t'$ is positive for all $t > t_1$ and we get

$$0 < a_t' \leq a_{t_1}' \prod_{s=t_1}^{t-1}\left(1 - \frac{q}{s+b}\right) = \mathcal{O}\left(\frac{1}{t^q}\right) = o\left(\frac{1}{t}\right).$$

In both cases, (A.5) is verified. □

**Lemma A.4.** *Let $x^\star$ be a regular solution of* (VI)*. Then, there exists constants $r, \alpha, \beta > 0$ such that $V$ is $\beta$-Lipschitz continuous on $\mathcal{K} := \mathbb{B}_r(x^\star)$ and $\langle V(x), x - x^\star \rangle \geq \alpha\|x - x^\star\|^2$ for all $x \in U := \mathcal{X} \cap \mathcal{K}$.*

*Proof.* The Lipschitz continuity is straightforward: a $C^1$-smooth operator is necessarily locally Lipschitz and thus Lipshitz on every compact. The proof consists in establishing the existence of $\alpha$. To this end, we consider the following function:

$$\phi: \quad \mathbb{R}^{d \times d} \quad \longrightarrow \quad \mathbb{R}$$
$$G \quad \longmapsto \quad \min_{z \in \mathrm{TC}_\mathcal{X}(x^\star), \|z\|=1} z^\top G z$$

where $\mathrm{TC}_\mathcal{X}(x^\star)$ denotes the tangent cone to $\mathcal{X}$ at $x^\star$. The function $\phi$ is concave as it is defined as a pointwise minimum over a set of linear functions. This in turn implies the continuity $\phi$ because every concave function is continous on the interior of its effective domain. The solution $x^\star$ being regular, we have $\phi(\mathrm{Jac}_V(x^\star)) > 0$. Combined with the continuity of $\mathrm{Jac}_V$ in a neighborhood of $x^\star$, we deduce the existence of $r, \alpha > 0$ such that $\phi(\mathrm{Jac}_V(x)) \geq \alpha$ for all $x \in \mathcal{K} = \mathbb{B}_r(x^\star)$. Now let $x \in U = \mathcal{X} \cap \mathcal{K}$. It holds:

$$V(x) - V(x^\star) = \left(\int_0^1 \mathrm{Jac}_V(x^\star + \lambda(x - x^\star))\, d\lambda\right)(x - x^\star).$$

Consequently, writing $z = x - x^\star \in \mathrm{TC}_\mathcal{X}(x^\star)$, $x_\lambda' = x^\star + \lambda(x - x^\star) \in \mathcal{K}$, we have

$$\langle V(x) - V(x^\star), x - x^\star \rangle = z^\top\left(\int_0^1 \mathrm{Jac}_V(x_\lambda')\, d\lambda\right)z$$

$$\geq \left(\int \phi(\mathrm{Jac}_V(x_\lambda'))\, d\lambda\right)\|z\|^2 \geq \alpha\|z\|^2 = \alpha\|x - x^\star\|^2.$$

Finally, since $x^\star$ is a solution of (VI), we have $\langle V(x^\star), x - x^\star \rangle \geq 0$ and

$$\langle V(x), x - x^\star \rangle \geq \langle V(x) - V(x^\star), x - x^\star \rangle \geq \alpha\|x - x^\star\|^2.$$

This ends the proof. □

# B Proofs for the deterministic setting

## B.1 Proof of Lemma 2

In the definition of $\mathrm{Err}_R$, instead of taking $\mathcal{X}_R = \mathcal{X} \cap \mathbb{B}_R(0)$ we consider $\mathcal{X}_R = \mathcal{X} \cap \mathbb{B}_R(X_1)$. Summing (3) over $s$ and rearranging the term leads to

$$\sum_{s=1}^{t} 2\lambda_s \langle V(X_{s+\frac{1}{2}}), X_{s+\frac{1}{2}} - p \rangle \leq \|X_1 - p\|^2 - \|X_{t+1} - p\|^2 + \mu_1 - \mu_{t+1} \leq \|X_1 - p\|^2 + \mu_1. \quad \text{(B.1)}$$

For any $p \in \mathcal{X}_R$, we have $\|X_1 - p\|^2 \leq R^2$, and by monoticity of $V$,

$$\langle V(p), X_{s+\frac{1}{2}} - p \rangle \leq \langle V(X_{s+\frac{1}{2}}), X_{s+\frac{1}{2}} - p \rangle.$$

In other words, for all $p \in \mathcal{X}_R$,

$$2 \sum_{s=1}^{t} \lambda_s \langle V(p), X_{s+\frac{1}{2}} - p \rangle \leq R^2 + \mu_1. \quad \text{(B.2)}$$

Dividing the two sides of (B.2) by $2\sum_{s=1}^{t} \lambda_s$ and maximizing over $p \in \mathcal{X}_R$ leads to the desired result.

## B.2 Proof of Theorem 1

To facilitate analysis and presentation of our results, (PEG) and (OG) are initialized with random $X_{\frac{1}{2}}$ and $X_1$ in $\mathcal{X}$ while for (RG) we start with $X_0$ and $X_{\frac{1}{2}}$. We are constrained to have different initial states in (RG) due to its specific formulation.

The theorem is immediate from Lemma 2 if we know that (3) is verified by the generated iterates for some $(\lambda_t)_{t \in \mathbb{N}}, (\mu_t)_{t \in \mathbb{N}} \in \mathbb{R}_+^{\mathbb{N}}$. Below, we show it separately for PEG, OG and RG under Assumption 2 and with $\gamma$ selected as per the theorem statement. Moreover, we have $(\lambda_t)_{t \in \mathbb{N}} \equiv \gamma$ and $\mu_1 \leq \|X_1 - X_{\frac{1}{2}}\|^2$ for all methods, hence the corresponding bound in our statement. The arguments used in the proof are inspired from [19, 26, 45] but we emphasize the relation between the analyses of these algorithms by putting forward the technical Lemma A.2.

**Past Extra-Gradient (PEG).** For $t \geq 1$, the second inequality of Lemma A.2 (b) applied to $(x, y_1, y_2, x_1^+, x_2^+, \mathcal{C}_1, \mathcal{C}_2) \leftarrow (X_t, \gamma V(X_{t-\frac{1}{2}}), \gamma V(X_{t+\frac{1}{2}}), X_{t+\frac{1}{2}}, X_{t+1}, \mathcal{X}, \mathcal{X})$ results in

$$\begin{aligned}
\|X_{t+1} - p\|^2 &\leq \|X_t - p\|^2 - 2\gamma \langle V(X_{t+\frac{1}{2}}), X_{t+\frac{1}{2}} - p \rangle \\
&\quad + \gamma^2 \|V(X_{t+\frac{1}{2}}) - V(X_{t-\frac{1}{2}})\|^2 - \|X_{t+\frac{1}{2}} - X_t\|^2 \\
&\leq \|X_t - p\|^2 - 2\gamma \langle V(X_{t+\frac{1}{2}}), X_{t+\frac{1}{2}} - p \rangle \\
&\quad + \gamma^2 \beta^2 \|X_{t+\frac{1}{2}} - X_{t-\frac{1}{2}}\|^2 - \|X_{t+\frac{1}{2}} - X_t\|^2
\end{aligned} \quad \text{(B.3)}$$

where we used the fact that $V$ is $\beta$-Lipschitz continuous for the second inequality.

Now, let us use Young's inequality $\|a + b\|^2 \leq 2\|a\|^2 + 2\|b\|^2$ to get

$$\|X_{t+\frac{1}{2}} - X_{t-\frac{1}{2}}\|^2 \leq 2\|X_{t+\frac{1}{2}} - X_t\|^2 + 2\|X_t - X_{t-\frac{1}{2}}\|^2 \quad \text{(B.4)}$$

and the non-expansiveness of the projection to get for any $t \geq 2$,

$$\|X_t - X_{t-\frac{1}{2}}\|^2 \leq \|X_{t-1} - \gamma V(X_{t-\frac{1}{2}}) - X_{t-1} + \gamma V(X_{t-\frac{3}{2}})\|^2 \leq \gamma^2 \beta^2 \|X_{t-\frac{1}{2}} - X_{t-\frac{3}{2}}\|^2. \quad \text{(B.5)}$$

Combining (B.4) and (B.5), we obtain

$$\begin{aligned}
\|X_{t+\frac{1}{2}} - X_{t-\frac{1}{2}}\|^2 &\leq 2\|X_{t+\frac{1}{2}} - X_t\|^2 + 2\gamma^2 \beta^2 \|X_{t-\frac{1}{2}} - X_{t-\frac{3}{2}}\|^2 \\
&\leq 2\|X_{t+\frac{1}{2}} - X_t\|^2 + \frac{1}{2}\|X_{t-\frac{1}{2}} - X_{t-\frac{3}{2}}\|^2,
\end{aligned} \quad \text{(B.6)}$$

where we used the fact that $\gamma \leq 1/(2\beta)$ in the last inequality; and in order to display a telescopic term, we reformulate (B.6) as

$$\|X_{t+\frac{1}{2}} - X_{t-\frac{1}{2}}\|^2 = 2\|X_{t+\frac{1}{2}} - X_{t-\frac{1}{2}}\|^2 - \|X_{t+\frac{1}{2}} - X_{t-\frac{1}{2}}\|^2$$
$$\leq 4\|X_{t+\frac{1}{2}} - X_t\|^2 + \|X_{t-\frac{1}{2}} - X_{t-\frac{3}{2}}\|^2 - \|X_{t+\frac{1}{2}} - X_{t-\frac{1}{2}}\|^2. \quad \text{(B.7)}$$

We now substitute (B.7) in (B.3) to get for all $t \geq 2$,

$$\|X_{t+1} - p\|^2 \leq \|X_t - p\|^2 - 2\gamma\langle V(X_{t+\frac{1}{2}}), X_{t+\frac{1}{2}} - p\rangle + (4\gamma^2\beta^2 - 1)\|X_{t+\frac{1}{2}} - X_t\|^2$$
$$+ \gamma^2\beta^2(\|X_{t-\frac{1}{2}} - X_{t-\frac{3}{2}}\|^2 - \|X_{t+\frac{1}{2}} - X_{t-\frac{1}{2}}\|^2)$$
$$\leq \|X_t - p\|^2 - 2\gamma\langle V(X_{t+\frac{1}{2}}), X_{t+\frac{1}{2}} - p\rangle$$
$$+ \gamma^2\beta^2(\|X_{t-\frac{1}{2}} - X_{t-\frac{3}{2}}\|^2 - \|X_{t+\frac{1}{2}} - X_{t-\frac{1}{2}}\|^2),$$

and thus (3) holds true for all $t \geq 2$ with $\lambda_t = \gamma$ and $\mu_t = \gamma^2\beta^2\|X_{t-\frac{1}{2}} - X_{t-\frac{3}{2}}\|^2$.

Finally, for $t = 1$, we have

$$\gamma^2\beta^2\|X_{\frac{3}{2}} - X_{\frac{1}{2}}\|^2 - \|X_{\frac{3}{2}} - X_1\|^2$$
$$\leq 4\gamma^2\beta^2\|X_{\frac{3}{2}} - X_1\|^2 + 4\gamma^2\beta^2\|X_1 - X_{\frac{1}{2}}\|^2 - \gamma^2\beta^2\|X_{\frac{3}{2}} - X_{\frac{1}{2}}\|^2 - \|X_{\frac{3}{2}} - X_1\|^2$$
$$\leq 4\gamma^2\beta^2\|X_1 - X_{\frac{1}{2}}\|^2 - \gamma^2\beta^2\|X_{\frac{3}{2}} - X_{\frac{1}{2}}\|^2,$$

which, plugged into (B.3) gives

$$\|X_2 - p\|^2 \leq \|X_1 - p\|^2 - 2\gamma\langle V(X_{\frac{3}{2}}), X_{\frac{3}{2}} - p\rangle + \gamma^2\beta^2\|X_{\frac{3}{2}} - X_{\frac{1}{2}}\|^2 - \|X_{\frac{3}{2}} - X_1\|^2$$
$$\leq \|X_1 - p\|^2 - 2\gamma\langle V(X_{\frac{3}{2}}), X_{\frac{3}{2}} - p\rangle + 4\gamma^2\beta^2\|X_1 - X_{\frac{1}{2}}\|^2 - \gamma^2\beta^2\|X_{\frac{3}{2}} - X_{\frac{1}{2}}\|^2 \quad \text{(B.8)}$$

which also matches (3) for $t = 1$ with $\lambda_t = \gamma$, $\mu_2$ as defined previously, and $\mu_1 = 4\gamma^2\beta^2\|X_1 - X_{\frac{1}{2}}\|^2 \leq \|X_1 - X_{\frac{1}{2}}\|^2$. Thus, Lemma 2 enables us to conclude the proof for Past Extra-Gradient (PEG).

**Optimistic Gradient (OG).**   The update of OG with constant step-size $\gamma$ can be written as

$$\begin{cases} X_{t+\frac{1}{2}} = \Pi_\mathcal{X}(X_t - \gamma V(X_{t-\frac{1}{2}})) \\ X_{t+1} = X_t - (X_t - X_{t+\frac{1}{2}} + \gamma V(X_{t+\frac{1}{2}}) - \gamma V(X_{t-\frac{1}{2}})) \end{cases}$$

In that form, we can use Lemma A.2 (a) with $(x, y_1, y_2, x_1^+, x_2^+, \mathcal{C}_1, \mathcal{C}_2) \leftarrow (X_t, \gamma V(X_{t-\frac{1}{2}}), X_t - X_{t+\frac{1}{2}} + \gamma V(X_{t+\frac{1}{2}}) - \gamma V(X_{t-\frac{1}{2}}), X_{t+\frac{1}{2}}, X_{t+1}, \mathcal{X}, \mathbb{R}^d)$ to get

$$\|X_{t+1} - p\|^2 = \|X_t - p\|^2 + \|X_{t+1} - X_{t+\frac{1}{2}}\|^2 - \|X_{t+\frac{1}{2}} - X_t\|^2$$
$$- 2\langle X_t - X_{t+\frac{1}{2}} + \gamma V(X_{t+\frac{1}{2}}) - \gamma V(X_{t-\frac{1}{2}}), X_{t+\frac{1}{2}} - p\rangle. \quad \text{(B.9)}$$

One the one hand, since $X_{t+\frac{1}{2}} = \Pi_\mathcal{X}(X_t - \gamma V(X_{t-\frac{1}{2}}))$ and $p \in \mathcal{X}$, we have

$$\langle X_{t+\frac{1}{2}} - (X_t - \gamma V(X_{t-\frac{1}{2}})), X_{t+\frac{1}{2}} - p\rangle \leq 0. \quad \text{(B.10)}$$

On the other other hand, by definition of $X_{t+1}$ and the $\beta$-Lipschitz continuity of $V$,

$$\|X_{t+1} - X_{t+\frac{1}{2}}\|^2 = \gamma^2\|V(X_{t+\frac{1}{2}}) - V(X_{t-\frac{1}{2}})\|^2 \leq \gamma^2\beta^2\|X_{t+\frac{1}{2}} - X_{t-\frac{1}{2}}\|^2. \quad \text{(B.11)}$$

Then, applying the same arguments used to get (B.7), we can show that for all $t \geq 2$,

$$\|X_{t+\frac{1}{2}} - X_{t-\frac{1}{2}}\|^2 \leq 4\|X_{t+\frac{1}{2}} - X_t\|^2 + \|X_{t-\frac{1}{2}} - X_{t-\frac{3}{2}}\|^2 - \|X_{t+\frac{1}{2}} - X_{t-\frac{1}{2}}\|^2. \quad \text{(B.12)}$$

Putting together (B.9), (B.10), (B.11), and (B.12), we obtain for $\gamma \leq 1/(2\beta)$ and for all $t \geq 2$,

$$\|X_{t+1} - p\|^2$$
$$\leq \|X_t - p\|^2 - 2\gamma\langle V(X_{t+\frac{1}{2}}), X_{t+\frac{1}{2}} - p\rangle + \gamma^2\beta^2\|X_{t+\frac{1}{2}} - X_{t-\frac{1}{2}}\|^2 - \|X_{t+\frac{1}{2}} - X_t\|^2$$
$$\leq \|X_t - p\|^2 - 2\gamma\langle V(X_{t+\frac{1}{2}}), X_{t+\frac{1}{2}} - p\rangle + \gamma^2\beta^2(\|X_{t-\frac{1}{2}} - X_{t-\frac{3}{2}}\|^2 - \|X_{t+\frac{1}{2}} - X_{t-\frac{1}{2}}\|^2).$$

Finally, since (B.8) is still true using the same argument as for PEG, (3) is satisfied by choosing the same $(\mu_t)_{t\in\mathbb{N}}$ and $(\lambda_t)_{t\in\mathbb{N}}$ as in the case of PEG; the same result thus holds for Optimistic Gradient (OG).

**Reflected Gradient (RG).** We recall the update rule of RG

$$\begin{cases} X_{t+\frac{1}{2}} = X_t - (X_{t-1} - X_t) \\ X_{t+1} = \Pi_{\mathcal{X}}(X_t - \gamma V(X_{t+\frac{1}{2}})). \end{cases}$$

As in the previous cases, we use Lemma A.2. Using the first inequality of Part (b) with $(x, y_1, y_2, x_1^+, x_2^+, \mathcal{C}_1, \mathcal{C}_2) \leftarrow (X_t, X_{t-1} - X_t, \gamma V(X_{t+\frac{1}{2}}), X_{t+\frac{1}{2}}, X_{t+1}, \mathbb{R}^d, \mathcal{X})$, we get

$$\|X_{t+1} - p\|^2 \leq \|X_t - p\|^2 + 2\langle \gamma V(X_{t+\frac{1}{2}}) - (X_{t-1} - X_t), X_{t+\frac{1}{2}} - X_{t+1}\rangle$$
$$- 2\gamma\langle V(X_{t+\frac{1}{2}}), X_{t+\frac{1}{2}} - p\rangle - \|X_{t+1} - X_{t+\frac{1}{2}}\|^2 - \|X_{t+\frac{1}{2}} - X_t\|^2. \quad (B.13)$$

As $X_t = \Pi_{\mathcal{X}}(X_{t-1} - \gamma V(X_{t-\frac{1}{2}}))$ and $X_{t-1}, X_{t+1} \in \mathcal{X}$, it follows

$$\langle X_t - (X_{t-1} - \gamma V(X_{t-\frac{1}{2}})), X_t - X_{t-1}\rangle \leq 0, \quad (B.14)$$
$$\langle X_t - (X_{t-1} - \gamma V(X_{t-\frac{1}{2}})), X_t - X_{t+1}\rangle \leq 0. \quad (B.15)$$

By summing (B.14) and (B.15) and rearranging the terms, we get

$$\langle X_t - X_{t-1}, X_{t+\frac{1}{2}} - X_{t+1}\rangle \leq -\langle \gamma V(X_{t-\frac{1}{2}}), X_{t+\frac{1}{2}} - X_{t+1}\rangle,$$

thus,

$$2\langle \gamma V(X_{t+\frac{1}{2}}) - (X_{t-1} - X_t), X_{t+\frac{1}{2}} - X_{t+1}\rangle$$
$$\leq 2\langle \gamma V(X_{t+\frac{1}{2}}) - \gamma V(X_{t-\frac{1}{2}}), X_{t+\frac{1}{2}} - X_{t+1}\rangle$$
$$\leq 2\gamma\beta\|X_{t+\frac{1}{2}} - X_{t-\frac{1}{2}}\|\|X_{t+\frac{1}{2}} - X_{t+1}\|. \quad (B.16)$$

Combining (B.13) and (B.16), we get

$$\|X_{t+1} - p\|^2 \leq \|X_t - p\|^2 + 2\gamma\beta\|X_{t+\frac{1}{2}} - X_{t-\frac{1}{2}}\|\|X_{t+\frac{1}{2}} - X_{t+1}\|$$
$$- 2\gamma\langle V(X_{t+\frac{1}{2}}), X_{t+\frac{1}{2}} - p\rangle - \|X_{t+1} - X_{t+\frac{1}{2}}\|^2 - \|X_{t+\frac{1}{2}} - X_t\|^2. \quad (B.17)$$

By using twice Young's inequality: i) $2\langle a, b\rangle \leq \varepsilon\|a\|^2 + (1/\varepsilon)\|b\|^2$ with $\varepsilon = 1/\sqrt{2}$; then ii) $\|a + b\|^2 \leq (1 + \varepsilon')\|a\|^2 + (1 + 1/\varepsilon')\|b\|^2$ with $\varepsilon' = 1 + \sqrt{2}$, we have

$$2\|X_{t+\frac{1}{2}} - X_{t-\frac{1}{2}}\|\|X_{t+\frac{1}{2}} - X_{t+1}\|$$
$$\leq \frac{1}{\sqrt{2}}\|X_{t+\frac{1}{2}} - X_{t-\frac{1}{2}}\|^2 + \sqrt{2}\|X_{t+\frac{1}{2}} - X_{t+1}\|^2$$
$$\leq (1 + \sqrt{2})\|X_{t+\frac{1}{2}} - X_t\|^2 + \|X_t - X_{t-\frac{1}{2}}\|^2 + \sqrt{2}\|X_{t+\frac{1}{2}} - X_{t+1}\|^2. \quad (B.18)$$

Substituting (B.18) into (B.17) yields

$$\|X_{t+1} - p\|^2 \leq \|X_t - p\|^2 - 2\gamma\langle V(X_{t+\frac{1}{2}}), X_{t+\frac{1}{2}} - p\rangle$$
$$+ ((1 + \sqrt{2})\gamma\beta - 1)\|X_{t+\frac{1}{2}} - X_t\|^2$$
$$+ \gamma\beta\|X_t - X_{t-\frac{1}{2}}\|^2 - (1 - \sqrt{2}\gamma\beta)\|X_{t+1} - X_{t+\frac{1}{2}}\|^2$$
$$\leq \|X_t - p\|^2 - 2\gamma\langle V(X_{t+\frac{1}{2}}), X_{t+\frac{1}{2}} - p\rangle$$
$$+ \gamma\beta\|X_t - X_{t-\frac{1}{2}}\|^2 - \gamma\beta\|X_{t+1} - X_{t+\frac{1}{2}}\|^2,$$

where in the last line, we used twice that $\gamma \leq 1/((1 + \sqrt{2})\beta)$. Once again, (3) is verified with the choice $\forall t \in \mathbb{N}, \mu_t = \gamma\beta\|X_t - X_{t-\frac{1}{2}}\|^2, \lambda_t = \gamma$ and the result thus holds for Reflected Gradient (RG). We also notice that $\mu_1 \leq \|X_1 - X_{\frac{1}{2}}\|^2$ since $\gamma\beta < 1$.

### B.3 Lemma 2 with other suboptimality measures

Here we discuss how the statement of Lemma 2, and consequently also that of Theorem 1, can be adjusted to consider more adapted convergence measures in the cases of loss minimization and min-max optimization. The notations are those of Examples 1 and 2, and we write $\bar{x} = (\sum_{s=1}^t \lambda_s)^{-1} \sum_{s=1}^t \lambda_s X_{s+\frac{1}{2}}$.

**Loss minimization.** $V = \nabla f$ is monotone implies the convexity of $f$, so

$$\langle V(X_{s+\frac{1}{2}}), X_{s+\frac{1}{2}} - p \rangle = \langle \nabla f(X_{s+\frac{1}{2}}), X_{s+\frac{1}{2}} - p \rangle \geq f(X_{s+\frac{1}{2}}) - f(p).$$

With Jensen's inequality we get,

$$\left( \sum_{s=1}^{t} \lambda_s \right)^{-1} \sum_{s=1}^{t} \lambda_s \langle V(X_{s+\frac{1}{2}}), X_{s+\frac{1}{2}} - p \rangle \geq \left( \sum_{s=1}^{t} \lambda_s \right)^{-1} \sum_{s=1}^{t} \lambda_s f(X_{s+\frac{1}{2}}) - f(p) \geq f(\bar{x}) - f(p)$$

This is true for any $p \in \mathcal{X}$, and especially for $p \in \mathcal{X}^\star$. Let $R = \mathrm{dist}(x_1, \mathcal{X}^\star)$. By invoking (B.1), we conclude

$$f(\bar{x}) - \min f \leq \frac{R^2 + \mu_1}{2 \sum_{s=1}^{t} \lambda_s}.$$

**Min-max optimization.** $V = (\nabla_\theta \mathcal{L}, -\nabla_\phi \mathcal{L})$ being monotone is equivalent to $\mathcal{L}$ being convex-concave. In such saddle-point problems, the quality of a candidate solution $\hat{x} = (\hat{\theta}, \hat{\phi})$ is often assessed via the *Nikaido–Isoda* function [37], defined here as

$$\mathrm{NI}(\hat{x}) = \sup_{\phi \in \Phi} \mathcal{L}(\hat{\theta}, \phi) - \inf_{\theta \in \Theta} \mathcal{L}(\theta, \hat{\phi}) \tag{NI}$$

provided of course that the right-hand side is well-posed. Its restricted variant $\mathrm{NI}_R$ can also be defined by analogy with the definition of $\mathrm{Err}_R$.

Let us denote $X_{s+\frac{1}{2}} = (\theta_{s+\frac{1}{2}}, \phi_{s+\frac{1}{2}})$ and $p = (\theta, \phi)$. By convex-concavity of $\mathcal{L}$, it holds

$$\begin{aligned}
\langle V(X_{s+\frac{1}{2}}), X_{s+\frac{1}{2}} - p \rangle &= \langle \nabla_\theta \mathcal{L}(\theta_{s+\frac{1}{2}}, \phi_{s+\frac{1}{2}}), \theta_{s+\frac{1}{2}} - \theta \rangle - \langle \nabla_\phi \mathcal{L}(\theta_{s+\frac{1}{2}}, \phi_{s+\frac{1}{2}}), \phi_{s+\frac{1}{2}} - \phi \rangle \\
&\geq \mathcal{L}(\theta_{s+\frac{1}{2}}, \phi_{s+\frac{1}{2}}) - \mathcal{L}(\theta, \phi_{s+\frac{1}{2}}) + \mathcal{L}(\theta_{s+\frac{1}{2}}, \phi) - \mathcal{L}(\theta_{s+\frac{1}{2}}, \phi_{s+\frac{1}{2}}) \\
&= \mathcal{L}(\theta_{s+\frac{1}{2}}, \phi) - \mathcal{L}(\theta, \phi_{s+\frac{1}{2}}).
\end{aligned}$$

We can again apply Jensen's inequality to show that

$$\left( \sum_{s=1}^{t} \lambda_s \right)^{-1} \sum_{s=1}^{t} \lambda_s \langle V(X_{s+\frac{1}{2}}), X_{s+\frac{1}{2}} - p \rangle \geq \mathcal{L}(\bar{\theta}, \phi) - \mathcal{L}(\theta, \bar{\phi}),$$

where we write $\bar{x} = (\bar{\theta}, \bar{\phi})$. By (B.1) and definition of the Nikaido–Isoda function, maximizing over $(\theta, \phi) \in \mathcal{X} \cap \mathbb{B}_R(X_1)$ gives

$$\mathrm{NI}_R(\bar{x}) \leq \frac{R^2 + \mu_1}{2 \sum_{s=1}^{t} \lambda_s}.$$

### B.4 Proof of Theorem 4

Here we provide a quick proof of Theorem 4. We do not try to optimize the constants and better results could be derived by examining each algorithm carefully. Note that since RG can evaluate $V$ at infeasible points, we need to strengthen condition (4) in Definition 3 to consider all $z$ in the *tangent span* of $\mathcal{X}$, i.e., the subspace of $\mathbb{R}^d$ spanned by all possible displacement vectors of the form $z = x' - x$, $x, x' \in \mathcal{X}$.

In order to show a local geometric convergence rate we only need to show that by choosing sufficiently small constant step-size and initializing at points sufficiently close to $x^\star$, we ensure $X_t \in \mathcal{K}$ for all $t \in \mathbb{N}/2$ (where $\mathcal{K}$ is defined in Lemma A.4 and this is in view of Theorem 2). In fact, although Theorem 2 is stated for strongly monotone operators, by carefully examining its proof, it turns out that we only need $\langle V(X_{t+\frac{1}{2}}), X_{t+\frac{1}{2}} - x^\star \rangle \geq \alpha \|X_{t+\frac{1}{2}} - x^\star\|^2$ for some constant $\alpha > 0$ and all $t \in \mathbb{N}$. We thus proceed to show that $\forall t \in \mathbb{N}/2, X_t \in \mathcal{K}$. To do so, let us show that one can choose the initial points and $\gamma$ so that $\forall t \in \mathbb{N}$, (i) $\|X_t - x^\star\|^2 \leq \frac{r^2}{4}$; (ii) $X_{t+\frac{1}{2}} \in \mathcal{K}$.

Part (i). It is proved in Appendix B.2 that the iterates of the 1-EG methods verify (3) under Assumption 2 (Lipschitz continuity) if $\gamma$ is smaller than some constant. By Lemma A.4 we know that $V$ is indeed Lipschitz continuous on the compact $\mathcal{K}$. Suppose that for all $s \in \mathbb{N}/2$, $s \leq t$, we have $X_s \in \mathcal{K}$, then it holds $\langle V(X_{s+\frac{1}{2}}), X_{s+\frac{1}{2}} - x^\star \rangle \geq 0$ for all $s \in \{1, ..., t-1\}$. This is true for PEG

and OG because $X_{s+\frac{1}{2}} \in \mathcal{X}$ and subsequently $X_{s+\frac{1}{2}} \in U = \mathcal{X} \cup \mathcal{K}$. For RG we did mention above that we need to relax the definition of a regular solution to consider all the $z \in \mathbb{R}^d$ and the statement of Lemma A.4 can also be modified accordingly. Using (3), we obtain[6]

$$\|X_t - x^\star\|^2 + \mu_t \leq \|X_1 - x^\star\|^2 + \mu_1.$$

for the three algorithms with $\mu_t \geq 0$. By imposing $X_{\frac{1}{2}} = X_1$ in PEG and OG, we get $\mu_1 = 0$. Similarly, we may impose $X_0 = X_{\frac{1}{2}}$ in RG, leading to $\mu_1 \leq \|X_1 - X_0\|^2 \leq \gamma^2 \|V(X_0)\|^2$. It is thus possible to choose the adequate initial points and $\gamma$ such that $\|X_1 - x^\star\|^2 + \mu_1 \leq \frac{r^2}{4}$, which in turn guarantees $\|X_t - x^\star\|^2 \leq \frac{r^2}{4}$.

Part (ii). We now proceed to prove that we may choose $\gamma$ sufficiently small such that if $\|X_t - x^\star\|^2 \leq \frac{r^2}{4}$ and $X_{t-\frac{1}{2}} \in \mathcal{K}$ then $X_{t+\frac{1}{2}} \in \mathcal{K}$. We notice that for the three algorithms, we have

$$\|X_{t+\frac{1}{2}} - X_t\|^2 \leq \gamma^2 \|V(X_{t-\frac{1}{2}})\|^2$$

by the non-expansiveness of the projection.[7] We define $M := \sup_{x \in \mathcal{K}} \|V(x)\| < \infty$ where the finiteness of $M$ comes from the continuity of $V$ and the boundedness of $\mathcal{K}$. We choose $\gamma \leq r/(2M)$ so that $\gamma^2 \|V(X_{t-\frac{1}{2}})\|^2 \leq \frac{r^2}{4}$ since $X_{t-\frac{1}{2}} \in \mathcal{K}$. Then, by Young's inequality, we get

$$\|X_{t+\frac{1}{2}} - x^\star\|^2 \leq 2\|X_{t+\frac{1}{2}} - X_t\|^2 + 2\|X_t - x^\star\|^2 \leq r^2.$$

In other words, $X_{t+\frac{1}{2}} \in \mathcal{K}$.

Conclusion. We first notice that the conditions on the initial points and the stepsize $\gamma$ do not depend on the iteration. Thus, by simple induction we have that if we initialize the algorithm such that

$$\gamma \leq r/(2M) \quad \text{and} \quad \|X_1 - x^\star\|^2 + \mu_1 \leq \frac{r^2}{4},$$

then for all $t \in \mathbb{N}/2$, $X_t \in \mathcal{K}$, concluding the proof.

## C   Proofs for the stochastic setting

Let us focus in this section on the (PEG) algorithm:

$$\begin{aligned} X_{t+\frac{1}{2}} &= \Pi_{\mathcal{X}}(X_t - \gamma_t V_{t-\frac{1}{2}}) \\ X_{t+1} &= \Pi_{\mathcal{X}}(X_t - \gamma_t V_{t+\frac{1}{2}}) \end{aligned} \tag{PEG}$$

Following Appendix B.2, we initialize the algorithm with random $X_{\frac{1}{2}}$ and $X_1$ in $\mathcal{X}$. Recall that $(\mathcal{F}_t)_{t \in \frac{\mathbb{N}}{2}}$ denotes the natural filtration associated with the sequence $(X_t)_{t \in \frac{\mathbb{N}}{2}}$. In the PEG algorithm, we have $\mathcal{F}_t = \mathcal{F}_{t+\frac{1}{2}}$ for all $t \in \mathbb{N}$ (thus $X_{t+\frac{1}{2}}$ is $\mathcal{F}_t$-measurable) so the zero-mean hypothesis (2a) can be written as $\mathbb{E}[Z_{t+\frac{1}{2}} \mid \mathcal{F}_t] = 0$.

### C.1   Proof of Theorem 5

**Last iterate convergence.**   As in the proof of Theorem 1, we first apply Lemma A.2 (b) with $(x, y_1, y_2, x_1^+, x_2^+, \mathcal{C}_1, \mathcal{C}_2) \leftarrow (X_t, \gamma_t V_{t-\frac{1}{2}}, \gamma_t V_{t+\frac{1}{2}}, X_{t+\frac{1}{2}}, X_{t+1}, \mathcal{X}, \mathcal{X})$ and the solution $x^\star \in \mathcal{X}$ as a trial point to obtain

$$\begin{aligned} \|X_{t+1} - x^\star\|^2 &\leq \|X_t - x^\star\|^2 - 2\gamma_t \langle V_{t+\frac{1}{2}}, X_{t+\frac{1}{2}} - x^\star \rangle \\ &\quad + \gamma_t^2 \|V_{t+\frac{1}{2}} - V_{t-\frac{1}{2}}\|^2 - \|X_{t+\frac{1}{2}} - X_t\|^2. \end{aligned} \tag{C.1}$$

The following holds true thanks to the law of total expectation,

$$\mathbb{E}[\|V_{t+\frac{1}{2}} - V_{t-\frac{1}{2}}\|^2]$$

$$\begin{aligned}
&= \mathbb{E}[\|V(X_{t+\frac{1}{2}}) - V_{t-\frac{1}{2}}\|^2 + 2\langle Z_{t+\frac{1}{2}}, V(X_{t+\frac{1}{2}}) - V_{t-\frac{1}{2}}\rangle + \|Z_{t+\frac{1}{2}}\|^2] \\
&= \mathbb{E}[\|V(X_{t+\frac{1}{2}}) - V_{t-\frac{1}{2}}\|^2] + 2\,\mathbb{E}[\mathbb{E}[\langle Z_{t+\frac{1}{2}}, V(X_{t+\frac{1}{2}}) - V_{t-\frac{1}{2}}\rangle \mid \mathcal{F}_t]] + \mathbb{E}[\|Z_{t+\frac{1}{2}}\|^2] \\
&= \mathbb{E}[\|V(X_{t+\frac{1}{2}}) - V_{t-\frac{1}{2}}\|^2] + \mathbb{E}[\|Z_{t+\frac{1}{2}}\|^2]. 
\end{aligned} \tag{C.2}$$

By Young's inequality, $\beta$-Lipschitz continuity of $V$, and non-expansiveness of the projection, we have

$$\begin{aligned}
\|V(X_{t+\frac{1}{2}}) - V_{t-\frac{1}{2}}\|^2 &\leq 2\|V(X_{t+\frac{1}{2}}) - V(X_{t-\frac{1}{2}})\|^2 + 2\|Z_{t-\frac{1}{2}}\|^2 \\
&\leq 2\beta^2\|X_{t+\frac{1}{2}} - X_{t-\frac{1}{2}}\|^2 + 2\|Z_{t-\frac{1}{2}}\|^2 \\
&\leq 4\beta^2\|X_{t+\frac{1}{2}} - X_t\|^2 + 4\beta^2\|X_t - X_{t-\frac{1}{2}}\|^2 + 2\|Z_{t-\frac{1}{2}}\|^2 \\
&\leq 4\beta^2\|X_{t+\frac{1}{2}} - X_t\|^2 + 4\gamma_{t-1}^2\beta^2\|V_{t-\frac{1}{2}} - V_{t-\frac{3}{2}}\|^2 + 2\|Z_{t-\frac{1}{2}}\|^2. 
\end{aligned} \tag{C.3}$$

Notice that the choice $b \geq 4\beta\gamma$ implies $8\gamma_t^2\beta^2 + 2\gamma_t\beta \leq 1$, which in turn yields $8\gamma_t^2\beta^2 \leq 1 - \alpha\gamma_t$. Combining (C.2) and (C.3), similarly to (B.7), we can thus show that

$$\begin{aligned}
\mathbb{E}[\|V_{t+\frac{1}{2}} - V_{t-\frac{1}{2}}\|^2] &\leq 8\beta^2\,\mathbb{E}[\|X_{t+\frac{1}{2}} - X_t\|^2] + 8\gamma_{t-1}^2\beta^2\,\mathbb{E}[\|V_{t-\frac{1}{2}} - V_{t-\frac{3}{2}}\|^2] \\
&\quad + 4\,\mathbb{E}[\|Z_{t-\frac{1}{2}}\|^2] + 2\,\mathbb{E}[\|Z_{t+\frac{1}{2}}\|^2] - \mathbb{E}[\|V_{t+\frac{1}{2}} - V_{t-\frac{1}{2}}\|^2] \\
&\leq 8\beta^2\,\mathbb{E}[\|X_{t+\frac{1}{2}} - X_t\|^2] + 6\sigma^2 \\
&\quad + \frac{\gamma_{t-1}^2}{\gamma_t^2}(1 - \alpha\gamma_t)\,\mathbb{E}[\|V_{t-\frac{1}{2}} - V_{t-\frac{3}{2}}\|^2] - \mathbb{E}[\|V_{t+\frac{1}{2}} - V_{t-\frac{1}{2}}\|^2], 
\end{aligned} \tag{C.4}$$

where in the last line we also use $\mathbb{E}[\|Z_{t-\frac{1}{2}}\|^2] \leq \sigma^2$, $\mathbb{E}[\|Z_{t+\frac{1}{2}}\|^2] \leq \sigma^2$.

We also have

$$\mathbb{E}[\langle V_{t+\frac{1}{2}}, X_{t+\frac{1}{2}} - x^\star\rangle] = \mathbb{E}[\mathbb{E}[\langle V_{t+\frac{1}{2}}, X_{t+\frac{1}{2}} - x^\star\rangle \mid \mathcal{F}_t]] = \mathbb{E}[\langle V(X_{t+\frac{1}{2}}), X_{t+\frac{1}{2}} - x^\star\rangle]. \tag{C.5}$$

Since $x^\star$ is the unique solution of (VI), it follows $\langle V(x^\star), X_{t+\frac{1}{2}} - x^\star\rangle \geq 0$. Consequently, with strong monotonicity of $V$, we get

$$\langle V(X_{t+\frac{1}{2}}), X_{t+\frac{1}{2}} - x^\star\rangle \geq \langle V(X_{t+\frac{1}{2}}) - V(x^\star), X_{t+\frac{1}{2}} - x^\star\rangle \geq \alpha\|X_{t+\frac{1}{2}} - x^\star\|^2.$$

By Young's inequality

$$\|X_t - x^\star\|^2 \leq 2\|X_t - X_{t+\frac{1}{2}}\|^2 + 2\|X_{t+\frac{1}{2}} - x^\star\|^2,$$

we can further write

$$\langle V(X_{t+\frac{1}{2}}), X_{t+\frac{1}{2}} - x^\star\rangle \geq \frac{\alpha}{2}\|X_t - x^\star\|^2 - \alpha\|X_t - X_{t+\frac{1}{2}}\|^2. \tag{C.6}$$

Taking expectation over (C.1) and using (C.4), (C.5), (C.6) leads to

$$\begin{aligned}
\mathbb{E}[\|X_{t+1} - x^\star\|^2] &\leq \mathbb{E}[\|X_t - x^\star\|^2] - \alpha\gamma_t\,\mathbb{E}[\|X_t - x^\star\|^2] + 2\alpha\gamma_t\,\mathbb{E}[\|X_t - X_{t+\frac{1}{2}}\|^2] \\
&\quad + \gamma_{t-1}^2(1 - \alpha\gamma_t)\,\mathbb{E}[\|V_{t-\frac{1}{2}} - V_{t-\frac{3}{2}}\|^2] \\
&\quad + 8\gamma_t^2\beta^2\,\mathbb{E}[\|X_{t+\frac{1}{2}} - X_t\|^2] - \gamma_t^2\,\mathbb{E}[\|V_{t+\frac{1}{2}} - V_{t-\frac{1}{2}}\|^2] \\
&\quad + 6\gamma_t^2\sigma^2 - \mathbb{E}[\|X_{t+\frac{1}{2}} - X_t\|^2] \\
&= (1 - \alpha\gamma_t)(\mathbb{E}[\|X_t - x^\star\|^2] + \gamma_{t-1}^2\,\mathbb{E}[\|V_{t-\frac{1}{2}} - V_{t-\frac{3}{2}}\|^2]) \\
&\quad + 6\gamma_t^2\sigma^2 - \gamma_t^2\,\mathbb{E}[\|V_{t+\frac{1}{2}} - V_{t-\frac{1}{2}}\|^2] \\
&\quad + (8\gamma_t^2\beta^2 + 2\alpha\gamma_t - 1)\,\mathbb{E}[\|X_{t+\frac{1}{2}} - X_t\|^2]. 
\end{aligned} \tag{C.7}$$

Using $8\gamma_t^2\beta^2 + 2\alpha\gamma_t - 1 \leq 0$, (C.7) reduces to

$$\begin{aligned}
\mathbb{E}[\|X_{t+1} - x^\star\|^2] &+ \gamma_t^2\,\mathbb{E}[\|V_{t+\frac{1}{2}} - V_{t-\frac{1}{2}}\|^2] \\
&\leq (1 - \alpha\gamma_t)(\mathbb{E}[\|X_t - x^\star\|^2] + \gamma_{t-1}^2\,\mathbb{E}[\|V_{t-\frac{1}{2}} - V_{t-\frac{3}{2}}\|^2]) + 6\gamma_t^2\sigma^2.
\end{aligned}$$

We conclude by applying Lemma A.3 with $a_t \leftarrow \mathbb{E}[\|X_t - x^\star\|^2] + \gamma_{t-1}^2 \mathbb{E}[\|V_{t-\frac{1}{2}} - V_{t-\frac{3}{2}}\|^2]$, $q \leftarrow \alpha\gamma$, $q' \leftarrow 6\gamma^2\sigma^2$, and $t_0 \leftarrow 2$, which gives

$$\mathbb{E}[\|X_t - x^\star\|^2] + \gamma_{t-1}^2 \mathbb{E}[\|V_{t-\frac{1}{2}} - V_{t-\frac{3}{2}}\|^2] \leq \frac{6\gamma^2\sigma^2}{\alpha\gamma - 1}\frac{1}{t} + o\left(\frac{1}{t}\right).$$

The second term on the left-hand side (LHS) of the inequality is always positive, and (5) follows immediately.

**Ergodic convergence.** The convergence of $\bar{X}_t$ as shown in (6) can be deduce directly from above by using Jensen's inequality:

$$\mathbb{E}[\|\bar{X}_t - x^\star\|^2] \leq \frac{1}{t}\sum_{s=1}^{t} \mathbb{E}[\|X_s - x^\star\|^2],$$

and then we bound the right-hand side (RHS) of the inequality by (5).

## C.2 Proof of Theorem 6

We start by defining some important quantities that will be used in our proof. For any $T \geq 1$, we set

$$S_T := \sum_{t=1}^{T} 2\gamma_t \langle Z_{t+\frac{1}{2}}, X_{t+\frac{1}{2}} - x^\star \rangle,$$

$$R_T := \sum_{t=1}^{T} 2\gamma_t^2 (\|V_{t+\frac{1}{2}}\|^2 + \|V_{t-\frac{1}{2}}\|^2),$$

$$Q_T := S_T^2 + R_T.$$

Notice that $S_T$, $R_T$ and $Q_T$ are not $\mathcal{F}_T$-measurable but $\mathcal{F}_{T+1}$-measurable (due to the terms in $Z_{T+\frac{1}{2}}$ and $V_{T+\frac{1}{2}}$). For the sake of simplicity, we also write $\xi_{t+\frac{1}{2}} := \langle Z_{t+\frac{1}{2}}, X_{t+\frac{1}{2}} - x^\star \rangle$ so that $S_T = \sum_{t=1}^{T} 2\gamma_t \xi_{t+\frac{1}{2}}$ and $\mathbb{E}[\xi_{t+\frac{1}{2}} \mid \mathcal{F}_t] = 0$.

Regarding the choice of $U$ and $U_1$, we invoke Lemma A.4 to obtain the corresponding $\alpha$, $r$ and $U$. We then set $U_1 := \mathcal{X} \cap \mathbb{B}_{r/4}(x^\star)$. Let us consider the following events for $T \geq 1$,

$$H_T := \left\{ \max_{1 \leq t \leq T} Q_t \leq \varepsilon := \min\left(\frac{r^2}{8}, \frac{r^4}{16}\right) \right\},$$

$$E_T := \left\{ \forall t \in \{1, ..., T\}, X_{t+\frac{1}{2}} \in U \right\}.$$

We additionally define $Q_0 := 2\gamma_1^2 \|V_{\frac{1}{2}}\|^2$, $H_0 := \{Q_0 \leq \varepsilon\}$ and $H_{-1} := E_0 := \Omega$, where $\Omega$ denotes the whole sample space. It follows from the definitions that both $(H_T)_{T\geq-1}$ and $(E_T)_{T\geq0}$ are decreasing sequences of events. Moreover, we have $H_T \in \mathcal{F}_{T+1}$ while $E_T \in \mathcal{F}_T$. Also notice that $E_\infty = \bigcap_{T\geq0} E_T$.

In terms of notation, for an event $E \subseteq \Omega$, we denote by $\mathbb{1}_E$ its indicator function and $E^c$ its complementary. For any pair of events $E, F \subseteq \Omega$, we denote by $E \setminus F$ the event "$E$ and not $F$" i.e., $E \cap F^c$.

The proof of the theorem relies on the two following lemmas.

**Lemma C.1.** *For any $T \geq 0$, we have the inclusion $H_{T-1} \subseteq E_T$.*

*Proof.* We prove this result by induction.

Initialization: $H_{-1} \subseteq E_0$ is clear. To prove that we also have $H_0 \subseteq E_1$, we use Young's inequality to get

$$\|X_{\frac{3}{2}} - x^\star\|^2 \leq 2\|X_{\frac{3}{2}} - X_1\|^2 + 2\|X_1 - x^\star\|^2. \tag{C.8}$$

On the one hand, since $X_1 \in U_1$ by assumption, it holds $\|X_1 - x^\star\|^2 \leq \frac{r^2}{16}$. On the other hand,

$$2\|X_{\frac{3}{2}} - X_1\|^2 = 2\|\Pi_{\mathcal{X}}(X_1 - \gamma_1 V_{\frac{1}{2}}) - \Pi_{\mathcal{X}}(X_1)\|^2 \leq 2\gamma_1\|V_{\frac{1}{2}}\|^2 = Q_0$$

For any realization in $H_0$, we have $2\gamma_1 \|V_{\frac{1}{2}}\|^2 \le \frac{r^2}{8}$; and so we can deduce from (C.8) that $\|X_{\frac{3}{2}} - x^\star\|^2 \le \frac{r^2}{4} < r^2$. Since $X_{\frac{3}{2}} \in \mathcal{X}$, it follows that $X_{\frac{3}{2}} \in U$. This means that $H_0 \subseteq E_1$.

Inductive step: Suppose that $H_{T-1} \subseteq E_T$ holds for some $T \ge 1$. We would like to prove $H_T \subseteq E_{T+1}$. To do so, we show that $\|X_{T+1} - x^\star\|^2 \le \frac{7}{16} r^2$ for any realization in $H_T$. Applying Lemma A.2 (b) as in (C.1) yields for all $t \in \{1, ..., T\}$,

$$\begin{aligned}
\|X_{t+1} - x^\star\|^2 &\le \|X_t - x^\star\|^2 - 2\gamma_t \langle V_{t+\frac{1}{2}}, X_{t+\frac{1}{2}} - x^\star \rangle \\
&\quad + \gamma_t^2 \|V_{t+\frac{1}{2}} - V_{t-\frac{1}{2}}\|^2 - \|X_{t+\frac{1}{2}} - X_t\|^2 \\
&\le \|X_t - x^\star\|^2 - 2\gamma_t \langle V(X_{t+\frac{1}{2}}), X_{t+\frac{1}{2}} - x^\star \rangle \\
&\quad - 2\gamma_t \langle Z_{t+\frac{1}{2}}, X_{t+\frac{1}{2}} - x^\star \rangle + 2\gamma_t^2 (\|V_{t+\frac{1}{2}}\|^2 + \|V_{t-\frac{1}{2}}\|^2) \\
&\le \|X_t - x^\star\|^2 - 2\gamma_t \xi_{t+\frac{1}{2}} + 2\gamma_t^2 (\|V_{t+\frac{1}{2}}\|^2 + \|V_{t-\frac{1}{2}}\|^2), \quad \text{(C.9)}
\end{aligned}$$

where in the last line we can use $\langle V(X_{t+\frac{1}{2}}), X_{t+\frac{1}{2}} - x^\star \rangle \ge 0$ since by induction hypothesis, $H_T \subseteq H_{T-1} \subseteq E_T$, which means for any realization in $H_T$, $X_{t+\frac{1}{2}} \in U$ for all $t \in \{1, ..., T\}$.

Summing (C.9) from $t = 1$ to $T$ gives

$$\begin{aligned}
\|X_{T+1} - x^\star\|^2 &\le \|X_1 - x^\star\|^2 - \sum_{t=1}^{T} 2\gamma_t \xi_{t+\frac{1}{2}} + \sum_{t=1}^{T} 2\gamma_t^2 (\|V_{t+\frac{1}{2}}\|^2 + \|V_{t-\frac{1}{2}}\|^2) \\
&= \|X_1 - x^\star\|^2 - S_T + R_T.
\end{aligned}$$

By definition of $H_T$, we have $S_T^2 \le Q_T \le \frac{r^4}{16}$ (so $|S_T| \le \frac{r^2}{4}$) and $R_T \le Q_T \le \frac{r^2}{8}$. Using that $\|X_1 - x^\star\|^2 \le \frac{r^2}{16}$ by assumption, it follows immediately that $\|X_{T+1} - x^\star\|^2 \le \frac{7}{16} r^2$.

Finally, in order to bound $\|X_{T+\frac{3}{2}} - x^\star\|^2$, we again rely on Young's inequality:

$$\begin{aligned}
\|X_{T+\frac{3}{2}} - x^\star\|^2 &\le 2\|X_{T+\frac{3}{2}} - X_{T+1}\|^2 + 2\|X_{T+1} - x^\star\|^2 \\
&\le 2\gamma_{T+1}^2 \|V_{T+\frac{1}{2}}\|^2 + 2\|X_{T+1} - x^\star\|^2. \quad \text{(C.10)}
\end{aligned}$$

For any realization in $H_T$, we have that

i) $\quad 2\gamma_{T+1}^2 \|V_{T+\frac{1}{2}}\|^2 \le 2\gamma_T^2 \|V_{T+\frac{1}{2}}\|^2 \le R_T \le Q_T \le \dfrac{r^2}{8}$;

ii) $\quad 2\|X_{T+1} - x^\star\|^2 \le \dfrac{7}{8} r^2.$

Thus, (C.10) implies that $\|X_{T+\frac{3}{2}} - x^\star\|^2 \le r^2$, and subsequently $X_{T+\frac{3}{2}} \in U$. As $H_T \subseteq E_T$ and $E_{T+1} = \{X_{T+\frac{3}{2}} \in U\} \cap E_T$, we have proven that $H_T \subseteq E_{T+1}$. $\qquad \square$

**Lemma C.2.** For $t \ge 1$, we have the following recurrence inequality

$$\mathbb{E}[Q_t \mathbb{1}_{H_{t-1}}] \le \mathbb{E}[Q_{t-1} \mathbb{1}_{H_{t-2}}] + \gamma_t^2 \mathcal{M} - \varepsilon \, \mathbb{P}(H_{t-2} \setminus H_{t-1}), \quad \text{(C.11)}$$

where $\mathcal{M} := 4M^2 + 4\sigma^2 + 4r^2\sigma^2$ and $\varepsilon := \min\left(\frac{r^2}{8}, \frac{r^4}{16}\right)$.

Moreover, if $t = 1$, the bound can be refined to

$$\mathbb{E}[Q_1 \mathbb{1}_{H_0}] \le \mathbb{E}[Q_0 \mathbb{1}_{H_{-1}}] + \gamma_1^2 (2M^2 + 2\sigma^2 + 4r^2\sigma^2) - \varepsilon \, \mathbb{P}(H_{-1} \setminus H_0). \quad \text{(C.12)}$$

*Proof.* We decompose

$$\begin{aligned}
\mathbb{E}[Q_t \mathbb{1}_{H_{t-1}}] &= \mathbb{E}[(Q_t - Q_{t-1}) \mathbb{1}_{H_{t-1}}] + \mathbb{E}[Q_{t-1} \mathbb{1}_{H_{t-1}}] \\
&= \mathbb{E}[(Q_t - Q_{t-1}) \mathbb{1}_{H_{t-1}}] + \mathbb{E}[Q_{t-1} \mathbb{1}_{H_{t-2}}] - \mathbb{E}[Q_{t-1} \mathbb{1}_{H_{t-2} \setminus H_{t-1}}], \quad \text{(C.13)}
\end{aligned}$$

where the second equality comes from the fact that as $H_{t-1} \subseteq H_{t-2}$, we have $H_{t-1} = H_{t-2} \setminus (H_{t-2} \setminus H_{t-1})$.

For $t \geq 2$, we write

$$
\begin{aligned}
Q_t &= S_t^2 + R_t \\
&= S_{t-1}^2 + 4\gamma_t \xi_{t+\frac{1}{2}} S_{t-1} + 4\gamma_t^2 \xi_{t+\frac{1}{2}}^2 + R_{t-1} + 2\gamma_t^2(\|V_{t+\frac{1}{2}}\|^2 + \|V_{t-\frac{1}{2}}\|^2) \\
&= Q_{t-1} + 4\gamma_t \xi_{t+\frac{1}{2}} S_{t-1} + 4\gamma_t^2 \xi_{t+\frac{1}{2}}^2 + 2\gamma_t^2(\|V_{t+\frac{1}{2}}\|^2 + \|V_{t-\frac{1}{2}}\|^2). \quad (C.14)
\end{aligned}
$$

Since $S_{t-1}$ and $H_{t-1}$ are $\mathcal{F}_t$-measurable, we get

$$
\mathbb{E}[\xi_{t+\frac{1}{2}} S_{t-1} \mathbb{1}_{H_{t-1}}] = \mathbb{E}[\mathbb{E}[\xi_{t+\frac{1}{2}} \mid \mathcal{F}_t] S_{t-1} \mathbb{1}_{H_{t-1}}] = 0. \quad (C.15)
$$

By Lemma C.1, $H_{t-1} \subseteq E_t$ which means that for any realization in $H_{t-1}$, we have $X_{t+\frac{1}{2}} \in U$. Therefore, $\|X_{t+\frac{1}{2}} - x^\star\|^2 \mathbb{1}_{H_{t-1}} \leq r^2 \mathbb{1}_{H_{t-1}}$ and consequently

$$
\begin{aligned}
\xi_{t+\frac{1}{2}}^2 \mathbb{1}_{H_{t-1}} &= \langle Z_{t+\frac{1}{2}}, X_{t+\frac{1}{2}} - x^\star \rangle^2 \mathbb{1}_{H_{t-1}} \\
&\leq \|Z_{t+\frac{1}{2}}\|^2 \|X_{t+\frac{1}{2}} - x^\star\|^2 \mathbb{1}_{H_{t-1}} \leq \|Z_{t+\frac{1}{2}}\|^2 r^2 \mathbb{1}_{H_{t-1}}.
\end{aligned}
$$

Using again that $H_{t-1}$ is $\mathcal{F}_t$-measurable along with the boundedness of the variance of $Z_{t+\frac{1}{2}}$ (see Eq. (2b)), we get

$$
\begin{aligned}
\mathbb{E}[\xi_{t+\frac{1}{2}}^2 \mathbb{1}_{H_{t-1}}] &\leq r^2 \mathbb{E}[\|Z_{t+\frac{1}{2}}\|^2 \mathbb{1}_{H_{t-1}}] = r^2 \mathbb{E}[\mathbb{E}[\|Z_{t+\frac{1}{2}}\|^2 \mid \mathcal{F}_t] \mathbb{1}_{H_{t-1}}] \\
&\leq r^2 \mathbb{E}[\sigma^2 \mathbb{1}_{H_{t-1}}] = r^2 \sigma^2 \mathbb{P}[H_{t-1}] \leq r^2 \sigma^2. \quad (C.16)
\end{aligned}
$$

Applying once again the techniques above and relying on the boundedness of $V$ (as for any realization in $H_{t-1} \subseteq E_t$ we have $X_{t+\frac{1}{2}} \in U$ and $M = \sup_{x \in U} V(x) < \infty$), we get

$$
\begin{aligned}
\mathbb{E}[\|V_{t+\frac{1}{2}}\|^2 \mathbb{1}_{H_{t-1}}] &= \mathbb{E}[(\|V(X_{t+\frac{1}{2}})\|^2 + 2\langle Z_{t+\frac{1}{2}}, V(X_{t+\frac{1}{2}})\rangle + \|Z_{t+\frac{1}{2}}\|^2) \mathbb{1}_{H_{t-1}}] \\
&= \mathbb{E}[\|V(X_{t+\frac{1}{2}})\|^2 \mathbb{1}_{H_{t-1}}] \\
&\quad + 2\mathbb{E}[\mathbb{E}[\langle Z_{t+\frac{1}{2}}, V(X_{t+\frac{1}{2}})\rangle \mid \mathcal{F}_t] \mathbb{1}_{H_{t-1}}] + \mathbb{E}[\|Z_{t+\frac{1}{2}}\|^2 \mathbb{1}_{H_{t-1}}] \\
&= \mathbb{E}[\|V(X_{t+\frac{1}{2}})\|^2 \mathbb{1}_{H_{t-1}}] + 0 + \mathbb{E}[\mathbb{E}[\|Z_{t+\frac{1}{2}}\|^2 \mid \mathcal{F}_t] \mathbb{1}_{H_{t-1}}] \\
&\leq M^2 + \sigma^2. \quad (C.17)
\end{aligned}
$$

Using that $H_{t-1} \subseteq H_{t-2}$ and repeating the arguments leading to (C.17), we have

$$
\mathbb{E}[\|V_{t-\frac{1}{2}}\|^2 \mathbb{1}_{H_{t-1}}] \leq \mathbb{E}[\|V_{t-\frac{1}{2}}\|^2 \mathbb{1}_{H_{t-2}}] \leq M^2 + \sigma^2. \quad (C.18)
$$

Combining (C.14), (C.15), (C.16), (C.17) and (C.18), we get

$$
\mathbb{E}[(Q_t - Q_{t-1}) \mathbb{1}_{H_{t-1}}] \leq \gamma_t^2(4M^2 + 4\sigma^2 + 4r^2\sigma^2) = \gamma_t^2 \mathcal{M}. \quad (C.19)
$$

For the last term on the RHS of (C.13), we get by definition that for any realization in $H_{t-2} \setminus H_{t-1}$, $Q_{t-1} > \varepsilon$ and thus

$$
\mathbb{E}[Q_{t-1} \mathbb{1}_{H_{t-2} \setminus H_{t-1}}] \geq \varepsilon \mathbb{E}[\mathbb{1}_{H_{t-2} \setminus H_{t-1}}] = \varepsilon \mathbb{P}(H_{t-2} \setminus H_{t-1}). \quad (C.20)
$$

Substituting (C.19) and (C.20) into (C.13) gives exactly (C.11).

The case $t = 1$ is proved similarly. In fact,

$$
Q_1 - Q_0 = 4\gamma_1^2 \xi_{\frac{3}{2}}^2 + 2\gamma_1^2 \|V_{\frac{3}{2}}\|^2.
$$

Consequently by using $H_0 \subseteq E_1$, we have

$$
\mathbb{E}[(Q_1 - Q_0) \mathbb{1}_{H_0}] \leq \gamma_1^2(2M^2 + 2\sigma^2 + 4r^2\sigma^2).
$$

By definition $H_{-1} \setminus H_0 = \{Q_0 > \varepsilon\}$, which shows (C.20) is equally true with $t = 1$. (C.12) can then be immediately deduced from (C.13). $\qquad \square$

*Proof of Theorem 6.*

(a) We first show that by choosing $b$ sufficiently large, we have $\mathbb{P}(H_T) \geq 1 - \delta$ for all $T \geq -1$ (when $T = -1$, $H_{-1} = \Omega$). To do so, we will work on the complementary event $H_T{}^c = H_{T-1}{}^c \cup (H_{T-1} \setminus H_T)$ and prove that $\mathbb{P}(H_T{}^c) \leq \delta$. We start by bounding $\mathbb{P}(H_{T-1} \setminus H_T)$,

$$
\begin{aligned}
\varepsilon \, \mathbb{P}(H_{T-1} \setminus H_T) &= \varepsilon \, \mathbb{P}(\{Q_T > \varepsilon\} \cap H_{T-1}) \\
&= \mathbb{E}[\varepsilon \, \mathbb{1}_{\{Q_T > \varepsilon\} \cap H_{T-1}}] \\
&\leq \mathbb{E}[Q_T \, \mathbb{1}_{\{Q_T > \varepsilon\} \cap H_{T-1}}] \\
&\leq \mathbb{E}[Q_T \, \mathbb{1}_{H_{T-1}}].
\end{aligned}
\tag{C.21}
$$

The last line is true since $Q_T$ is a positive random variable.

We now use Lemma C.2 by summing (C.11) from $t = 2$ to $T$ and (C.12) which leads to

$$
\mathbb{E}[Q_T \, \mathbb{1}_{H_{T-1}}] \leq \mathbb{E}[Q_1 \, \mathbb{1}_{H_0}] + \sum_{t=2}^{T} \gamma_t^2 \mathcal{M} - \sum_{t=2}^{T} \varepsilon \, \mathbb{P}(H_{t-2} \setminus H_{t-1})
$$

$$
\leq \mathbb{E}[Q_0 \, \mathbb{1}_{H_{-1}}] + \gamma_1^2 (2M^2 + 2\sigma^2 + 4r^2\sigma^2) + \sum_{t=2}^{T} \gamma_t^2 \mathcal{M} - \sum_{t=1}^{T} \varepsilon \, \mathbb{P}(H_{t-2} \setminus H_{t-1})
$$

$$
= \mathbb{E}[Q_0] + \gamma_1^2 (2M^2 + 2\sigma^2 + 4r^2\sigma^2) + \sum_{t=2}^{T} \gamma_t^2 \mathcal{M} - \varepsilon \, \mathbb{P}(H_{T-1}{}^c),
\tag{C.22}
$$

where in the last line we use that $H_{-1} = \Omega$ and $H_{T-1}{}^c = H_{-1} \setminus H_{T-1} = \dot{\bigcup}_{1 \leq t \leq T}(H_{t-2} \setminus H_{t-1})$ ( with $\dot{\bigcup}$ denoting the disjoint union) to get that $\mathbb{P}(H_{T-1}{}^c) = \sum_{t=1}^{T} \mathbb{P}(H_{t-2} \setminus H_{t-1})$. Since we initialize with $X_{\frac{1}{2}} \in U$, we have

$$
\mathbb{E}[Q_0] = 2\gamma_1^2 \, \mathbb{E}[\|V_{\frac{1}{2}}\|^2] \leq 2\gamma_1^2 (M^2 + \sigma^2).
\tag{C.23}
$$

We set $\Gamma := \sum_{t=1}^{\infty} \gamma_t^2 < \infty$. Combining (C.21), (C.22) and (C.23), we obtain

$$
\begin{aligned}
\mathbb{P}(H_T{}^c) &= \mathbb{P}(H_{T-1} \setminus H_T) + \mathbb{P}(H_{T-1}{}^c) \\
&\leq \frac{1}{\varepsilon} \, \mathbb{E}[Q_T \, \mathbb{1}_{H_{T-1}}] + \mathbb{P}(H_{T-1}{}^c) \\
&\leq \frac{1}{\varepsilon} \sum_{t=1}^{T} \gamma_t^2 \mathcal{M} - \mathbb{P}(H_{T-1}{}^c) + \mathbb{P}(H_{T-1}{}^c) \leq \frac{\Gamma \mathcal{M}}{\varepsilon}.
\end{aligned}
$$

As $\Gamma$ converges to $0$ when $b \to \infty$, for any $\delta > 0$ one can choose $b$ sufficiently large so that $\Gamma \leq \delta \varepsilon / \mathcal{M}$; we then have $\mathbb{P}(H_T{}^c) \leq \delta$, or equivalently, $\mathbb{P}(H_T) \geq 1 - \delta$ for all $T \geq -1$.

Since $H_{T-1} \subseteq E_T$ from Lemma C.1, we know that by choosing $b$ sufficiently large, we have $\mathbb{P}(E_T) \geq \mathbb{P}(H_{T-1}) \geq 1 - \delta$ for all $T \geq 0$. As $(E_T)_{T \geq 1}$ is a decreasing sequence of events and $E_\infty = \bigcap_{T \geq 0} E_T$, by continuity from above we have

$$
\mathbb{P}(E_\infty) = \lim_{T \to \infty} \mathbb{P}(E_T) \geq 1 - \delta,
$$

concluding the proof.

(b) Applying Lemma A.2 (b) gives

$$
\begin{aligned}
\|X_{t+1} - x^\star\|^2 &\leq \|X_t - x^\star\|^2 - 2\gamma_t \langle V_{t+\frac{1}{2}}, X_{t+\frac{1}{2}} - x^\star \rangle \\
&\quad + \gamma_t^2 \|V_{t+\frac{1}{2}} - V_{t-\frac{1}{2}}\|^2 - \|X_{t+\frac{1}{2}} - X_t\|^2 \\
&\leq \|X_t - x^\star\|^2 - 2\gamma_t \langle V(X_{t+\frac{1}{2}}), X_{t+\frac{1}{2}} - x^\star \rangle \\
&\quad - 2\gamma_t \langle Z_{t+\frac{1}{2}}, X_{t+\frac{1}{2}} - x^\star \rangle \\
&\quad + 2\gamma_t^2 (\|V_{t+\frac{1}{2}}\|^2 + \|V_{t-\frac{1}{2}}\|^2) - \|X_{t+\frac{1}{2}} - X_t\|^2.
\end{aligned}
$$

Furthermore, for any realization in $E_t$, $X_{t+\frac{1}{2}} \in U$ so that $\langle V(X_{t+\frac{1}{2}}), X_{t+\frac{1}{2}} - x^\star \rangle \geq \alpha \|X_{t+\frac{1}{2}} - x^\star\|^2$ and thus equation (C.6) holds, which allows us to write

$$
\|X_{t+1} - x^\star\|^2 \mathbb{1}_{E_t} \leq \|X_t - x^\star\|^2 \mathbb{1}_{E_t} - 2\gamma_t \langle V(X_{t+\frac{1}{2}}), X_{t+\frac{1}{2}} - x^\star \rangle \mathbb{1}_{E_t}
$$

$$- 2\gamma_t \langle Z_{t+\frac{1}{2}}, X_{t+\frac{1}{2}} - x^\star \rangle \mathbb{1}_{E_t}$$
$$+ 2\gamma_t^2 (\|V_{t+\frac{1}{2}}\|^2 + \|V_{t-\frac{1}{2}}\|^2)\, \mathbb{1}_{E_t} - \|X_{t+\frac{1}{2}} - X_t\|^2\, \mathbb{1}_{E_t}$$
$$\leq (1 - \alpha\gamma_t)\|X_t - x^\star\|^2\, \mathbb{1}_{E_t} - 2\gamma_t \langle Z_{t+\frac{1}{2}}, X_{t+\frac{1}{2}} - x^\star \rangle \mathbb{1}_{E_t}$$
$$+ 2\gamma_t^2 (\|V_{t+\frac{1}{2}}\|^2 + \|V_{t-\frac{1}{2}}\|^2)\, \mathbb{1}_{E_t} + (2\alpha\gamma_t - 1)\|X_{t+\frac{1}{2}} - X_t\|^2\, \mathbb{1}_{E_t}. \quad \text{(C.24)}$$

Similarly to (C.17) and (C.18), we have

$$\mathbb{E}[\|V_{t+\frac{1}{2}}\|^2\, \mathbb{1}_{E_t}] \leq M^2 + \sigma^2,$$
$$\mathbb{E}[\|V_{t-\frac{1}{2}}\|^2\, \mathbb{1}_{E_t}] \leq \mathbb{E}[\|V_{t-\frac{1}{2}}\|^2\, \mathbb{1}_{E_{t-1}}] \leq M^2 + \sigma^2.$$

We also recall that as $E_t \in \mathcal{F}_t$, it holds

$$\mathbb{E}[\langle Z_{t+\frac{1}{2}}, X_{t+\frac{1}{2}} - x^\star \rangle \mathbb{1}_{E_t}] = \mathbb{E}[\mathbb{E}[\langle Z_{t+\frac{1}{2}}, X_{t+\frac{1}{2}} - x^\star \rangle \mid \mathcal{F}_t]\, \mathbb{1}_{E_t}] = 0.$$

Taking expectation over (C.24) then leads to

$$\mathbb{E}[\|X_{t+1} - x^\star\|^2\, \mathbb{1}_{E_t}] \leq (1 - \alpha\gamma_t)\, \mathbb{E}[\|X_t - x^\star\|^2\, \mathbb{1}_{E_t}]$$
$$+ 4\gamma_t^2 (M^2 + \sigma^2) + (2\alpha\gamma_t - 1)\, \mathbb{E}[\|X_{t+\frac{1}{2}} - X_t\|^2\, \mathbb{1}_{E_t}].$$

We can choose $b$ sufficiently large so that $2\alpha\gamma_t - 1 \leq 0$ for all $t \geq 1$. Using $E_t \subseteq E_{t-1}$, we obtain

$$\mathbb{E}[\|X_{t+1} - x^\star\|^2\, \mathbb{1}_{E_t}] \leq (1 - \alpha\gamma_t)\, \mathbb{E}[\|X_t - x^\star\|^2\, \mathbb{1}_{E_{t-1}}] + 4\gamma_t^2 (M^2 + \sigma^2).$$

By applying Lemma A.3 with $a_t \leftarrow \mathbb{E}[\|X_t - x^\star\|^2\, \mathbb{1}_{E_{t-1}}]$, $q \leftarrow \alpha\gamma$, $q' \leftarrow 4\gamma^2(M^2 + \sigma^2)$, and $t_0 \leftarrow 1$, we get

$$\mathbb{E}[\|X_t - x^\star\|^2\, \mathbb{1}_{E_{t-1}}] \leq \frac{4\gamma^2(M^2 + \sigma^2)}{\alpha\gamma - 1} \frac{1}{t} + o\left(\frac{1}{t}\right).$$

Finally,

$$\mathbb{E}[\|X_t - x^\star\|^2 \mid E_\infty] = \frac{\mathbb{E}[\|X_t - x^\star\|^2\, \mathbb{1}_{E_\infty}]}{\mathbb{P}(E_\infty)}$$
$$\leq \frac{\mathbb{E}[\|X_t - x^\star\|^2\, \mathbb{1}_{E_{t-1}}]}{1 - \delta}$$
$$\leq \frac{4\gamma^2(M^2 + \sigma^2)}{(\alpha\gamma - 1)(1 - \delta)} \frac{1}{t} + o\left(\frac{1}{t}\right)$$

and our proof is complete. $\qquad \square$

*Remark.* We notice that to complete the above proof, we only require Eq. (2a) and Eq. (2b) to be held on the event $\{X_{t+\frac{1}{2}} \in U\}$. For example, in (C.16) we want $\mathbb{E}[\mathbb{E}[\|Z_{t+\frac{1}{2}}\|^2 \mid \mathcal{F}_t]\, \mathbb{1}_{H_{t-1}}] \leq \sigma^2$ which is true if Eq. (2b) holds on $\{X_{t+\frac{1}{2}} \in U\}$ since $H_{t-1} \subseteq E_t \subseteq \{X_{t+\frac{1}{2}} \in U\}$. This assumption is much weaker and more sensible. It in particular shows that to obtain local guarantee we indeed only need the noise to be bounded locally.