[Reviews · NeurIPS 2019]

Reviewer 1



The paper is well-written and can be easily followed. I did not carefully check the proofs, but the results are reasonable and the methodology is correct. The results are mildly significant in my opinion.

Reviewer 2



This paper solves an open problem in the analysis of some variants of extragradient. On the positive side, I found the paper to be clear and well written. I particularly appreciated the review of single-call variants of extragradient in Section 3. I reviewed some key results that are proven in the appendix (Lemma 2), which seemed correct. On the negative side, I found their results on non-monotone operators (Theorem 4.) to be rather disappointing, since it only applies when the iterates are already in a neighborhood of the solution and the operator to be (basically) strongly monotone in that region. It is nothing more than a localized version of previous results. # Post rebuttal I read te autors rebuttal and the other reviewer's review. My score remains unchanged.

Reviewer 3



In this paper, the authors study the EG algorithm and its variants in the optimization of VI with single oracle. In the convergence analysis, a unified way is proposed. Then the ergodic and last-iterate convergences are discussed with monotone and non-monotone operators in both deterministic and stochastic cases. The authors mentioned their motivation clearly. Some related works were discussed in detail and the paper is well written.

[Author Response · NeurIPS 2019]

First off, we would like to thank to the reviewers for their careful reading of our manuscript and their positive evaluation. We address their detailed remarks below:

**Reviewer 1.**

1. Regarding experiments with generative adversarial networks (GANs): Given the wide variety of single-call extra-gradient proxies, we found that a proper assessment and evaluation of the benefits of GAN training with one variant or another would take us too far afield relative to the scope of this paper. Instead, we chose to focus on synthetic experiments where the algorithms' convergence properties can be illustrated and validated directly.

2. On the constant $M$ in Theorem 6: In general non-monotone problems, $M$ is a local bound on the norm of $V$, so it does not have a deleterious effect on the algorithm's actual convergence time. In particular, if $x^\star$ is interior (as is typically the case in many machine learning models), continuity of $V$ implies that $M$ is small if $U$ is also chosen to be small.

3. On the error function Err: This "gap function" is the standard figure of merit for measuring the quality of a candidate solution in general variational inequalities; for instance, the celebrated $\mathcal{O}(1/T)$ convergence rate of Nemirovski's [1] mirror-prox algorithm – and, later, Nesterov's [2] dual extrapolation scheme – is established relative to Err. By a straightforward modification of Lemma 2, it is trivial to transform a convergence guarantee relative to Err to a value convergence guarantee (when the operator is the gradient of a loss function to be minimized), a bound on the Nikaido-Isoda function (for saddle-point problems), or the squared norm distance (if the operator is strongly monotone). We will be happy to discuss all this in more detail in a revision!

**Reviewer 2.**   Regarding the assumptions of Theorem 4 (local convergence in deterministic VIs): By necessity, local convergence results rely on the local structure of the operator and, in non-pathological cases, this is fully captured by the operator's Jacobian at the point in question (or higher-order derivates for more singular cases). Without an assumption of this kind, it does not seem possible to establish local attraction (or, rather, asymptotic stability) under gradient-based methods. We must also stress here that Theorem 4 essentially serves as a starting point and comparison baseline for Theorem 6 that investigates the convergence rate in stochastic environments. Although it may be possible to relax this assumption (e.g., for cases where there is a stable manifold of solutions), we chose to keep a straightforward and easy to parse hypothesis for clarity and readability.

**Reviewer 3.**   On the use of the terms "large enough" and "sufficiently small":

- In Theorem 1, our use of the term was an oversight, the bound for $\text{Err}_R$ holds for all $R > 0$. [At the same time, by Lemma 1, $\mathcal{X}_R$ should contain a solution of (VI) for $\text{Err}_R$ to be a meaningful performance measure]

- In Theorem 4, the exact choice of $\gamma$ may differ from one variant to another; in Theorem 6, we need $\gamma > 1/\alpha$ but the requirement for $b$ is considerably more tedious to write down (and not particularly informative to boot). We chose to omit the detailed expressions and descriptions in the statements of the theorems in order to streamline our presentation and to avoid interrupting the flow of our discussion.

We will of course be happy to make these choices explicit in the supplement and to add a series of remarks pointing the readers to the relevant discussion in the supplement.

# References

[1] Nemirovski, Arkadi Semen. 2004. Prox-method with rate of convergence $O(1/t)$ for variational inequalities with Lipschitz continuous monotone operators and smooth convex-concave saddle point problems. *SIAM Journal on Optimization* **15**(1) 229–251.

[2] Nesterov, Yurii. 2007. Dual extrapolation and its applications to solving variational inequalities and related problems. *Mathematical Programming* **109**(2) 319–344.


[Meta-Review · NeurIPS 2019]

This paper presents a unified analysis of three well known extra gradient methods that require only one gradient query for each step, in deterministic and stochastic settings obtaining the tight O(1/t) convergence rate.